

# Genome-wide characterization and expression analysis of *PP2CA* family members in response to ABA and osmotic stress in *Gossypium*

Tingting Lu[1,2,*], Gaofeng Zhang[1,*], Yibin Wang[1], Shibin He[1], Lirong Sun[1] and Fushun Hao[1]

[1] State Key Laboratory of Cotton Biology, Henan Key Laboratory of Plant Stress Biology, School of Life Sciences, Henan University, Kaifeng, Henan, China
[2] Henan University of Animal Husbandry and Economy, Zhengzhou, Henan, China
* These authors contributed equally to this work.

## ABSTRACT

Clade A type 2C protein phosphatases (PP2CAs), as central regulators of abscisic acid (ABA) signaling, negative control growth, development and responses to multiple stresses in plants. PP2CA gene families have been characterized at genome-wide levels in several diploid plants like *Arabidopsis* and rice. However, the information about genome organization, phylogenesis and putative functions of PP2CAs in *Gossypium* is lacking. Here, PP2CA family members were comprehensively analyzed in four *Gossypium* species including the diploid progenitor *Gossypium arboreum*, *G. raimondii* and the tetraploid *G. hirsutum* and *G. barbadense*, and 14, 13, 27, and 23 PP2CA genes were identified in the genomic sequences of these plants, respectively. Analysis results showed that most *Gossypium* PP2CAs were highly conserved in chromosomal locations, structures, and phylogeny among the four cotton species. Segmental duplication might play important roles in the formation of the *PP2CAs*, and most PP2CAs may be under purifying selection in *Gossypium* during evolution. The majority of the *PP2CAs* were expressed specifically in diverse tissues, and highly expressed in flowers in *G. hirsutum*. The *GhPP2CAs* displayed diverse expression patterns in responding to ABA and osmotic stress. Yeast-two hybrid assays revealed that many GhPP2CAs were capable of interaction with the cotton ABA receptors pyrabactin resistance1/PYR1-like/regulatory components of ABA receptors (PYR1/PYL/RCAR) GhPYL2-2D (Gh_D08G2587), GhPYL6-2A (Gh_A06G1418), and GhPYL9-2A (Gh_A11G0870) in the presence and/or absence of ABA. These results gave a comprehensive view of the *Gossypium* PP2CAs and are valuable for further studying the functions of PP2CAs in *Gossypium*.

# INTRODUCTION

Protein phosphorylation and dephosphorylation, as two central mechanisms of cellular signal transduction, play pivotal roles in many biological processes including growth,

Corresponding authors
Lirong Sun, sunlr9208@henu.edu.cn
Fushun Hao, haofsh@henu.edu.cn

development, and adaptations to various environmental stimuli in plants (*Schweighofer, Hirt & Meskiene, 2004*). They are catalyzed by protein kinases and phosphatases, respectively. Phosphatases are generally categorized into serine/threonine (Ser/Thr) phosphatases and tyrosine phosphatases according to the different amino acid residues they dephosphorylate. Based on biochemical and pharmacological properties, Ser/Thr phosphatases can be further classified into three large families: phosphoprotein phosphatases (PPs), phosphoprotein metallophosphatases, and aspartate-based protein phosphatases (*Schweighofer, Hirt & Meskiene, 2004*; *Kerk, Templeton & Moorhead, 2007*; *Fuchs et al., 2013*; *Singh et al., 2015*). The PPs includes PP1, PP2A, PP2B, PP4, PP5, PP6, and PP7, the phosphoprotein metallophosphatases consist of $Mg^{2+}/Mn^{2+}$-dependent type 2C protein phosphatases (PP2Cs) and other $Mg^{2+}$-dependent phosphatases (*Schweighofer, Hirt & Meskiene, 2004*; *Singh et al., 2010*, *2015*; *Fuchs et al., 2013*). PP2Cs, which play key roles in dephosphorylation events in plants, belong to a large subfamily, and can be further divided into 11 clades (A–K) in *Arabidopsis* and rice (*Singh et al., 2010*) and 12 clades (A–L) in *Brachypodium distachyon* (*Cao et al., 2016*). Among these, Clade A proteins of PP2Cs (PP2CAs) are the ones of well-studied PP2Cs in *Arabidopsis*, and they have been shown to have important roles in controlling abscisic acid (ABA) signaling, and negatively regulate plant growth, development and response to various biotic and abiotic stresses in plants (*Tähtiharju & Palva, 2001*; *Fuchs et al., 2013*; *Singh et al., 2015*). In *Arabidopsis* genome, nine *PP2CA* members have been identified. They are *ABI1* (*ABA insensitive 1*), *ABI2*, *HAB1* (*Homology to ABI1*), *HAB2*, *AHG1* (*ABA hypersensitive germination 1*), *HAI1* (*Highly ABA-induced PP2C1*), *HAI2*, *HAI3*, and *AHG3/AtPP2CA* (*Fuchs et al., 2013*). These genes, particularly *ABI1*, *ABI2*, and *AHG3/PP2CA* alone or cooperatively control ABA-mediated transpiration, stomatal closure, seed germination and root growth, and are involved in the regulation of many abiotic stress responses like drought, high salinity, cold, heat, and potassium deprivation (*Schweighofer, Hirt & Meskiene, 2004*; *Rubio et al., 2009*; *Singh et al., 2015*). Some *PP2CAs* also play important roles in responses to pathogen attack (*Schweighofer, Hirt & Meskiene, 2004*; *Singh et al., 2015*). *PP2CAs* are functionally redundant, and their expression is upregulated by high concentrations of ABA (*Rubio et al., 2009*; *Singh et al., 2015*). Moreover, PP2CAs physically interact with numerous cytosolic and nuclear localized proteins such as AtHB6 (Homeobox protein 6), CIPK8 (Calcineurin B–like protein-interacting protein kinase 8), CIPK24, and SnRK2s (Sucrose nonfermenting one-related protein kinases subfamily two proteins) (*Ohta et al., 2003*; *Fuchs et al., 2013*; *Singh et al., 2015*). SnRK2s exert central and positive roles in ABA signal cascade in plants (*Fujii & Zhu, 2009*; *Fujii et al., 2009*).

Recently, ABA receptors pyrabactin resistance1/PYR1-like/regulatory components of ABA receptors (PYR1/PYL/RCAR) (named PYLs for simplicity) have been found (*Ma et al., 2009*; *Park et al., 2009*). This is a breathtaking discovery in plants. PP2CAs were identified as coreceptors, specifically interact with PYLs and control ABA signaling. In the presence of ABA, ABA binds to PYLs, further interacts with and inhibits the activities of PP2CAs; thereby releasing and activating SnRK2s. SnRK2s subsequently regulate

multiple downstream transcriptional factors and other proteins to trigger ABA responses (*Fujii et al., 2009*; *Geiger et al., 2009*; *Lee et al., 2009*).

PP2C gene families including *PP2CAs* have been analyzed at genome-wide levels in *Arabidopsis*, rice, maize and *B. distachyon* (*Xue et al., 2008*; *Fujita et al., 2009*; *Wei & Pan, 2014*; *Cao et al., 2016*). The domain structure of PP2CAs was also studied (*Schweighofer, Hirt & Meskiene, 2004*). Moreover, the expression patterns of *PP2CAs* have been examined in response to ABA and multiple stresses in *Arabidopsis*, rice, maize and *B. distachyon* (*Xue et al., 2008*; *Wei & Pan, 2014*; *Zhang et al., 2017a*). However, genomic information and expression profiles of PP2CAs in cotton is unknown to date.

Cotton is the most important fiber crop which provides the spinnable lint for the textile industry in the world. The yield and quality of cotton are adversely affected by many abiotic stresses such as drought and high salinity, which are governed by ABA signaling (*Hauser, Waadt & Schroeder, 2011*; *Liang et al., 2017*; *Ullah et al., 2017*). Therefore, it is essential for us to uncover the functional mechanisms of PP2CAs in ABA signal transduction pathway in cotton. Here, we carried out a genome-wide identification of PP2CA gene family in diploid *Gossypium arboreum* (A2) and *G. raimondii* (D5), and their descendant tetraploid species *G. hirsutum* (AD1) and *G. barbadense* (AD2). The evolutionary relationships of these PP2CAs were analyzed. Changes in the transcriptional levels of the *PP2CAs* were also investigated in diverse tissues and in response to ABA and osmotic stress in *G. hirsutum*. Furthermore, the interactions between *G. hirsutum* PP2CAs and several GhPYLs were detected by the yeast-two hybrid method. These results may be valuable for further functional characterization of cotton PP2CAs in ABA signaling in the future.

## MATERIALS AND METHODS

### Analysis of the PP2C family in four *Gossypium* species

To explore all the members of the PP2C family in *Gossypium*, the protein sequences of 80 AtPP2Cs were initially applied as queries to search against the databases of *G. arboreum* (BGI-CGB v2.0 assembly genome), *G. raimondii* (JGI assembly v2.0 data), *G. hirsutum* (NAU-NBI v1.1 assembly genome) (www.cottongen.org), and *G. barbadense* (http://database.chgc.sh.cn/cotton/index.html), respectively, using the BLAST program with default setting (*E*-value $< e^{-10}$) (*Camacho et al., 2009*). After removing the redundant sequences from the data set, the putative *Gossypium* PP2Cs were then characterized using the PP2C model (PF00481) (http://pfam.xfam.org/) by the Hmmer software (http://hmmer.org/), and the proteins without a PP2C catalytic domain were deleted. The molecular weight (MW) and isoelectric point of the *Gossypium* PP2Cs were predicted by the online tool ExPaSy (http://web.expasy.org/protparam/), which can give various physico-chemical properties of a protein based on its amino acid sequence (the extinction coefficient and the absorbance of a native protein in water at 280 nm were used). The composition and position of exons and introns of the PP2CAs were obtained from CottonGen (https://www.cottongen.org/) and characterized by the Gene Structure Display Server tools (http://gsds.cbi.pku.edu.cn/) (*Hu et al., 2015*). The conserved domains of PP2CAs were validated in NCBI (https://www.ncbi.nlm.nih.gov/Structure/cdd/wrpsb.cgi)

using the automatic mode (*Marchler-Bauer et al., 2017*). The MEME program (meme-suite.org/tools/meme) was applied to determine the motifs of PP2CAs in *Gossypium* ("any number of repetitions" to be distributed in sequences was set). The locations of *Gossypium PP2CAs* in chromosomes were assessed using MapInspect software (http://www.mybiosoftware.com/mapinspect-compare-display-linkage-maps.html).

## Analysis of synteny and *Ka/Ks* of *PP2CAs*

The homologous regions of *PP2CAs* in *Gossypium* were identified by MCScanx software (http://chibba.pgml.uga.edu/mcscan2/) and syntenic blocks were determined by the CIRCOS program (http://www.circos.ca/). The syntenic maps of the *PP2CAs* were obtained using the circos-0.69±3 software with default parameters (http://www.circos.ca/). Some genes located within the same or adjacent intergenic region were regarded as tandem duplications. The nucleotide substitution parameters *Ka* (nonsynonymous) and *Ks* (synonymous) were assessed by the PAML program (http://abacus.gene.ucl.ac.uk/software/paml.html). Then, the ratio of *Ka/Ks* was calculated. *Ka/Ks* <1 means purifying selection; *Ka/Ks* = 1 indicates neutral selection, while *Ka/Ks* >1 represents positive selection (*Hurst, 2002*).

## Phylogenetic analysis of *PP2CAs*

The PP2CA databases were downloaded for *Arabidopsis thaliana* (http://www.arabidopsis.org/), *Theobroma cacao* (http://cocoagendb.cirad.fr), *Ricinus communis* (http://castorbean.jcvi.org), *Populus trichocarpa* (http://www.phytozome.net/poplar), *Glycine max* (http://www.phytozome.net/soybean), *B. distachyon* (http://plants.ensembl.org/Brachypodium_distachyon/Info/Index), *Oryza sativa* (http://rapdb.dna.affrc.go.jp), and the four *Gossypium* species mentioned above. The amino acid sequences of PP2CAs were aligned by the MUSCLE software (*Edgar, 2004*), and a phylogenetic tree of the PP2CAs was generated using the IQ-TREE server (http://www.iqtree.org/) following the maximum likelihood (ML) method (*Nguyen et al., 2015*; *Trifinopoulos et al., 2016*). The best-fitting model was chosen using the ModelFinder (*Kalyaanamoorthy et al., 2017*) and the support was assessed using the ultrafast bootstrap (*Minh, Nguyen & Von Haeseler, 2013*). Evolutionary tree was visualized using the FigTree v1.4.4 software (available from https://github.com/rambaut/figtree/releases).

## Measurements of *GhPP2CAs* expression in tissues and in response to ABA or osmotic stress

For measuring the expression of *GhPP2CAs* in tissues in each experiment, seeds of *G. hirsutum* L. acc. Texas Marker-1 (TM-1) were sown in pots containing the mixed nutrient soil (rich soil:vermiculite = 2:1, v/v) in a growth chamber. After 21 days, about two gram samples of roots, stems, or leaves were collected from 10 plants. About 20 flowers were collected 1 day post anthesis, and about five gram fibers were obtained from ovules 23 days post anthesis of cotton plants grown in the fields. For monitoring the expression of *GhPP2CAs* after ABA treatment or under osmotic stress, TM-1 seeds were germinated and planted in liquid 1/2 MS medium (*Murashige & Skoog, 1962*) in a growth chamber

(the medium was aerated. day/night temperature cycle of 28/26 °C, 14 h light/10 h dark, and about 50% relative humidity). Three weeks later, the plants were sprayed with 100 μM ABA or treated with 10% PEG6000 (dissolved in medium) for 0, 3, 6, 12, and 24 h, respectively. Then, about two gram roots were sampled, frozen in liquid nitrogen and stored at −70 °C. Total RNA was extracted from some of the samples and cDNA was generated according to the method described previously (*Ma et al., 2012*; *Zhang et al., 2017b*).

Quantitative real-time RT-PCR (qRT-PCR) experiments were constructed in an ABI 7,500 real-time PCR amplifier using the cDNA, SYBR Green Master mix, the specific primers of *GhPP2CA* genes (Table S1). *GhUBQ7* was used as an internal control (*Lu et al., 2017*). Experiments were independently repeated three times. The interval between two repeated experiments was 7–10 days.

### Monitoring protein interaction by yeast-two hybrid method

The CDS sequences of *GhPYLs* (*GhPYL2-2D*, *GhPYL6-2A*, and *GhPYL9-2A*) and *GhPP2CAs* were amplified, and cloned into pGADT7 and pGBKT7 vectors, respectively, using gene specific primers (Table S2). After sequencing, the fused vectors were transformed into AH109. The cotransformants were plated on nonselective SD/-Leu/-Trp solid medium and selective SD/-Leu/-Trp/-His/-Ade solid medium as described previously (*Lu et al., 2017*; *Zhang et al., 2017b*).

## RESULTS

### Genome-wide analysis of *PP2CAs* in four *Gossypium* species

To identify the putative PP2CA family members in *Gossypium*, the amino acid sequences of 80 *Arabidopsis* PP2Cs (*Xue et al., 2008*) were used to survey the *Gossypium* databases. Putative *PP2Cs* were assigned to a total of 114, 116, 239, and 232 genomic sequences that were retrieved from *G. arboreum*, *G. raimondii*, *G. hirsutum*, and *G. barbadense*, respectively. They were individually denominated as GaPP2Cs, GrPP2Cs, GhPP2Cs, and GbPP2Cs (Table S3). According to the phylogenetic relationships of PP2Cs between *Gossypium* and *Arabidopsis*, the *Gossypium* PP2Cs could be clustered into 12 clades (A–L) (Figs. S1–S4). The PP2CAs possessed 14 GaPP2CAs, 13 GrPP2CAs, 27 GhPP2CAs, and 23 GbPP2CAs, respectively. They were named individually according to their gene identifiers (Table 1). In this report, we focused on the PP2CA family members in the four *Gossypium* species.

It was found that the predicted coded amino acid lengths of *Gossypium* PP2CAs ranged from 118 to 593, with an average of 420. These PP2CAs had MWs of 12.8 kDa (GhPP2CA27) to 66 kDa (GaPP2CA6). The mean theoretical pIs of PP2CAs were approximately 5.9 with a minimum of 4.65 (GrPP2CA5) and a maximum of 8.74 (GhPP2CA22) (Table 1).

### Phylogenetic and structural analysis of PP2CAs in *Gossypium*

In order to understand the evolutionary relationship among GaPP2CAs, GrPP2CAs, GhPP2CAs, and GbPP2CAs, we conducted a phylogenetic tree using the protein sequences of the *Gossypium* PP2CAs (Fig. 1A). As expected, most of GaPP2CAs were individually

| Table 1 PP2CA family genes in *Gossypium*. | | | | |
|---|---|---|---|---|
| **Gene identifier** | **Gene name** | **Size (aa)** | **Mass (kDa)** | **pI** |
| Cotton_A_00941 | GaPP2CA1 | 420 | 46.9 | 6.61 |
| Cotton_A_01365 | GaPP2CA2 | 419 | 45.9 | 6.39 |
| Cotton_A_02676 | GaPP2CA3 | 494 | 53.4 | 5 |
| Cotton_A_03112 | GaPP2CA4 | 400 | 43.4 | 5.12 |
| Cotton_A_05714 | GaPP2CA5 | 470 | 51.8 | 5.82 |
| Cotton_A_11854 | GaPP2CA6 | 593 | 66 | 5.51 |
| Cotton_A_12895 | GaPP2CA7 | 416 | 45.4 | 6.07 |
| Cotton_A_14028 | GaPP2CA8 | 573 | 62.4 | 4.69 |
| Cotton_A_18842 | GaPP2CA9 | 393 | 43 | 5.25 |
| Cotton_A_25088 | GaPP2CA10 | 413 | 45.1 | 5.15 |
| Cotton_A_25089 | GaPP2CA11 | 413 | 45.1 | 5.15 |
| Cotton_A_27758 | GaPP2CA12 | 390 | 43.3 | 8.34 |
| Cotton_A_31719 | GaPP2CA13 | 413 | 45.4 | 5.55 |
| Cotton_A_34085 | GaPP2CA14 | 242 | 26.6 | 5.73 |
| Gh_A03G0373 | GhPP2CA1 | 416 | 45.9 | 8.09 |
| Gh_A05G0308 | GhPP2CA2 | 558 | 60.9 | 4.69 |
| Gh_A05G0782 | GhPP2CA3 | 416 | 45.4 | 6.07 |
| Gh_A05G1136 | GhPP2CA4 | 494 | 53.5 | 5.09 |
| Gh_A05G3030 | GhPP2CA5 | 413 | 45.4 | 5.52 |
| Gh_A06G0579 | GhPP2CA6 | 416 | 45.9 | 5.64 |
| Gh_A07G0123 | GhPP2CA7 | 471 | 52 | 5.58 |
| Gh_A08G2192 | GhPP2CA8 | 463 | 51.4 | 7.56 |
| Gh_A10G0578 | GhPP2CA9 | 397 | 42.8 | 5.76 |
| Gh_A10G1998 | GhPP2CA10 | 409 | 44.6 | 5.1 |
| Gh_A12G2380 | GhPP2CA11 | 420 | 46 | 6.27 |
| Gh_A13G0184 | GhPP2CA12 | 179 | 20 | 5.71 |
| Gh_A13G1741 | GhPP2CA13 | 400 | 43.3 | 5.12 |
| Gh_D03G1169 | GhPP2CA14 | 258 | 28.1 | 8.6 |
| Gh_D04G0612 | GhPP2CA15 | 413 | 45.5 | 5.61 |
| Gh_D05G0410 | GhPP2CA16 | 558 | 60.8 | 4.69 |
| Gh_D05G1309 | GhPP2CA17 | 494 | 53.5 | 5 |
| Gh_D05G3907 | GhPP2CA18 | 417 | 45.4 | 5.83 |
| Gh_D06G0657 | GhPP2CA19 | 393 | 43 | 5.49 |
| Gh_D07G2383 | GhPP2CA20 | 471 | 51.9 | 5.91 |
| Gh_D08G2557 | GhPP2CA21 | 463 | 51.3 | 6.48 |
| Gh_D10G0622 | GhPP2CA22 | 342 | 38.1 | 8.74 |
| Gh_D10G2305 | GhPP2CA23 | 411 | 45 | 5.05 |
| Gh_D12G2508 | GhPP2CA24 | 418 | 45.8 | 6.42 |
| Gh_D13G0199 | GhPP2CA25 | 416 | 46.2 | 6.53 |
| Gh_D13G2089 | GhPP2CA26 | 400 | 43.4 | 5.17 |
| Gh_Sca051315G01 | GhPP2CA27 | 118 | 12.8 | 4.93 |
| GOBAR_AA03316 | GbPP2CA1 | 413 | 45.4 | 5.52 |

| Gene identifier | Gene name | Size (aa) | Mass (kDa) | pI |
| --- | --- | --- | --- | --- |
| GOBAR_AA08348 | GbPP2CA2 | 409 | 44.6 | 5.1 |
| GOBAR_AA12929 | GbPP2CA3 | 416 | 45.8 | 8.07 |
| GOBAR_AA17958 | GbPP2CA4 | 393 | 43 | 5.31 |
| GOBAR_AA26179 | GbPP2CA5 | 418 | 45.9 | 6.39 |
| GOBAR_AA27223 | GbPP2CA6 | 456 | 50.6 | 7.16 |
| GOBAR_AA30591 | GbPP2CA7 | 470 | 51.8 | 5.58 |
| GOBAR_AA32894 | GbPP2CA8 | 558 | 60.9 | 4.67 |
| GOBAR_AA34839 | GbPP2CA9 | 416 | 45.4 | 6.07 |
| GOBAR_AA37246 | GbPP2CA10 | 558 | 60.8 | 4.69 |
| GOBAR_DD04210 | GbPP2CA11 | 416 | 46.1 | 6.76 |
| GOBAR_DD07153 | GbPP2CA12 | 371 | 40.7 | 6.07 |
| GOBAR_DD08720 | GbPP2CA13 | 392 | 43.2 | 7.2 |
| GOBAR_DD10426 | GbPP2CA14 | 385 | 42.4 | 6.64 |
| GOBAR_DD12855 | GbPP2CA15 | 302 | 33.4 | 6.46 |
| GOBAR_DD17767 | GbPP2CA16 | 389 | 42 | 4.96 |
| GOBAR_DD22444 | GbPP2CA17 | 400 | 43.3 | 5.11 |
| GOBAR_DD29723 | GbPP2CA18 | 408 | 44.9 | 8.52 |
| GOBAR_DD30544 | GbPP2CA19 | 411 | 45 | 5.05 |
| GOBAR_DD32331 | GbPP2CA20 | 226 | 25.2 | 5.58 |
| GOBAR_DD35228 | GbPP2CA21 | 470 | 51.8 | 5.82 |
| GOBAR_DD37768 | GbPP2CA22 | 470 | 51.8 | 5.82 |
| GOBAR_DD38067 | GbPP2CA23 | 494 | 53.5 | 5 |
| Gorai.001G013500 | GrPP2CA1 | 471 | 51.9 | 5.91 |
| Gorai.003G128900 | GrPP2CA2 | 416 | 45.9 | 8.09 |
| Gorai.004G284400 | GrPP2CA3 | 463 | 51.2 | 6.34 |
| Gorai.008G282600 | GrPP2CA4 | 414 | 45.4 | 6.3 |
| Gorai.009G042600 | GrPP2CA5 | 558 | 60.8 | 4.65 |
| Gorai.009G096200 | GrPP2CA6 | 416 | 45.4 | 5.73 |
| Gorai.009G143300 | GrPP2CA7 | 494 | 53.4 | 5.04 |
| Gorai.010G076700 | GrPP2CA8 | 393 | 43 | 5.48 |
| Gorai.011G071200 | GrPP2CA9 | 347 | 38.8 | 8.43 |
| Gorai.011G268200 | GrPP2CA10 | 411 | 44.9 | 5 |
| Gorai.012G072200 | GrPP2CA11 | 409 | 44.9 | 5.7 |
| Gorai.013G022100 | GrPP2CA12 | 415 | 46.1 | 6.2 |
| Gorai.013G230300 | GrPP2CA13 | 400 | 43.3 | 5 |

clustered closely with their corresponding orthologs of GhPP2CAs (GhPP2CA1–13) and GbPP2CAs (GbPP2CA1–10) in A genomes, and a majority of GrPP2CAs individually clustered closely with their homologs of GhPP2CAs (GhPP2CA14–27) and GbPP2CAs (GbPP2CA11–23) in D genomes. Noteworthily, GaPP2CA10 clustered together with GaPP2CA11, and a similar case occurred between GbPP2CA21/GbPP2CA22 and GbPP2CA16/GbPP2CA23. Moreover, homologues of 14 GaPP2CAs and of 13 GrPP2CAs

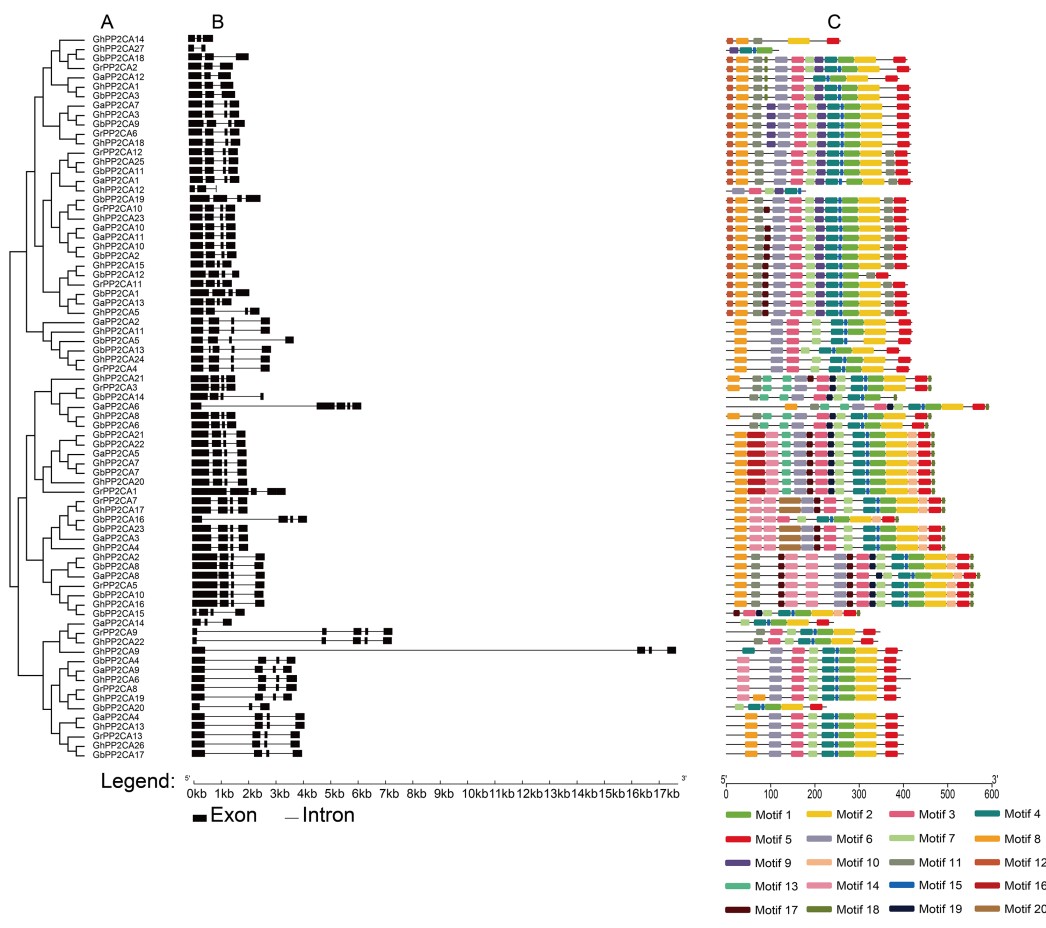

**Figure 1 Phylogenetic relationships, gene structures, and conserved motifs of PP2CA genes in *Gossypium*.** (A) The phylogenetic tree was generated by the maximum likelihood (ML) method, with ultrafast bootstrap. (B) Exon/intron architectures of *Gossypium* PP2C genes. The black boxes indicate exons, and the black lines represent introns. The sizes of exons and introns were determined by the scale at the bottom. (C) Distributions of conserved motifs. The motifs are displayed by 20 different color boxes.

were found in the *G. hirsutum* At and Dt subgenomes, respectively; and homologs of 10 GaPP2CAs (except GaPP2CA1, GaPP2CA3, GaPP2CA4, and GaPP2CA14) and 11 GrPP2CAs (except GrPP2CA6 and GrPP2CA9) were detected in the *G. barbadense* At′ or Dt′ subgenomes, respectively. Additionally, three pairs of paralogues with high sequence similarity in GbPP2CAs including GbPP2CA8/GbPP2CA10, GbPP2CA16/GbPP2CA23, GbPP2CA21/GbPP2CA22 were clustered together. They were seemingly derived from GaPP2CA8, GrPP2CA7, and GrPP2CA1, respectively.

Most *PP2CAs* had three to four exons except that *GaPP2CA6, GrPP2CA9, GhPP2CA22, GbPP2CA13* possessed five exons, and *GhPP2CA27* had two exons. Among the PP2CA genes, *GaPP2CA6, GrPP2CA9, GhPP2CA9*, and *GhPP2CA22* individually had a longer intron sequence than other genes did (Fig. 1B). These results indicate that the exon/intron structures of the *Gossypium* PP2CA genes were highly conserved.

The motif compositions of the PP2CA proteins were analyzed in the four *Gossypium* species. Twenty putative motifs named motif 1 to motif 20 were identified. Among those,

motif 1, 2, 3, 4, 5, 6, and 7 existed in every cluster and the majority of the PP2CA members. Moreover, most orthologous PP2CA proteins in the four *Gossypium* plants had the same or very similar compositions and distributions of motifs, suggesting that the PP2CA members in the same cluster likely share similar functions (Fig. 1C).

## Chromosomal distributions of *PP2CAs* in *Gossypium*

To determine the putative evolutionary relationships of the *Gossypium* PP2CA genes, the positions of the genes on chromosomes were analyzed. We found that the distributions of these *PP2CAs* were uneven. The 14 *GaPP2CAs*, 13 *GrPP2CAs*, 27 *GhPP2CAs*, and 23 *GbPP2CAs* were distributed on 8, 9, 16, and 14 chromosomes, respectively. Most of the chromosomes contained one PP2CA gene. By contrast, some chromosomes individually had two PP2CA genes. They were D11 and D13 in *G. raimondii*, At10, At13, Dt05, Dt10, and Dt13 in *G. hirsutum*, and Dt′07 and Dt′13 in *G. barbadense*. Besides, each of D09 and Dt′05 owned three PP2CAs. A13, At05, and At′05 separately possessed four *PP2CAs*. In contrast, *GaPP2CA5*, *GaPP2CA8*, *GaPP2CA14*, *GhPP2CA18*, *GhPP2CA20*, *GhPP2CA27*, *GbPP2CA2*, and *GbPP2CA12* were located on scaffolds, in which contigs were not spliced into any chromosome in genomic mapping.

We compared the positions of the orthologs of *GaPP2CAs*, *GrPP2CAs*, and *GhPP2CAs* or *GbPP2CAs* in chromosomes. As expected, most homologs of *GaPP2CAs* and *GrPP2CAs* in *G. hirsutum* were located in their corresponding At subgenomes and Dt subgenomes. A similar situation also occurred in *G. barbadense.* However, homologous genes of *GaPP2CAs* and *GrPP2CAs* were barely located in their corresponding homoeologous chromosomes and collinear loci in *G. hirsutum* and *G. barbadense* (Fig. 2). For example, GaPP2CA7 was found in chromosome A09, however, its ortholog GhPP2CA3 and GbPP2CA9 were present in chromosome At05 and At′05, respectively. These results imply that specific, unique, and complex variation events in *PP2CA*-contained homoeologous chromosomes may happen within each of the two diploid and tetraploid species during genetic evolution.

## Synteny analysis of *PP2CA* genes

During evolutionary processes, tandem and segmental duplications contribute to expanding gene family in plants (*Cannon et al., 2004*). We examined the duplication relationship of the *PP2CAs* among *G. arboreum*, *G. raimondii*, and *G. hirsutum* (the related database for *G. barbadense* was lacking). It was found that *GaPP2CA10* and *GaPP2CA11* joined together, and *GbPP2CA16* and *GbPP2CA23* clustered together in the chromosome. There are less than five genes between each pair of the genes, suggesting that the two pairs of genes are tandemly duplicated.

The synteny relationship of gene pairs was also explored among *GaPP2CAs*, *GrPP2CAs*, and *GhPP2CAs.* A total of 136 homologous gene pairs were observed in 133 collinearity blocks. Most of the blocks had one gene pair. Some blocks owned two gene pairs (*GrPP2CA9/GhPP2CA22*, *GrPP2CA10/GhPP2CA23*) between chromosomal D11 and Dt10. Another block harbored three gene pairs (*GrPP2CA5/GhPP2CA2*, *GrPP2CA6/GhPP2CA3*, *GrPP2CA7/GhPP2CA4*) between chromosomal D09 and At05 (Fig. 3). These findings

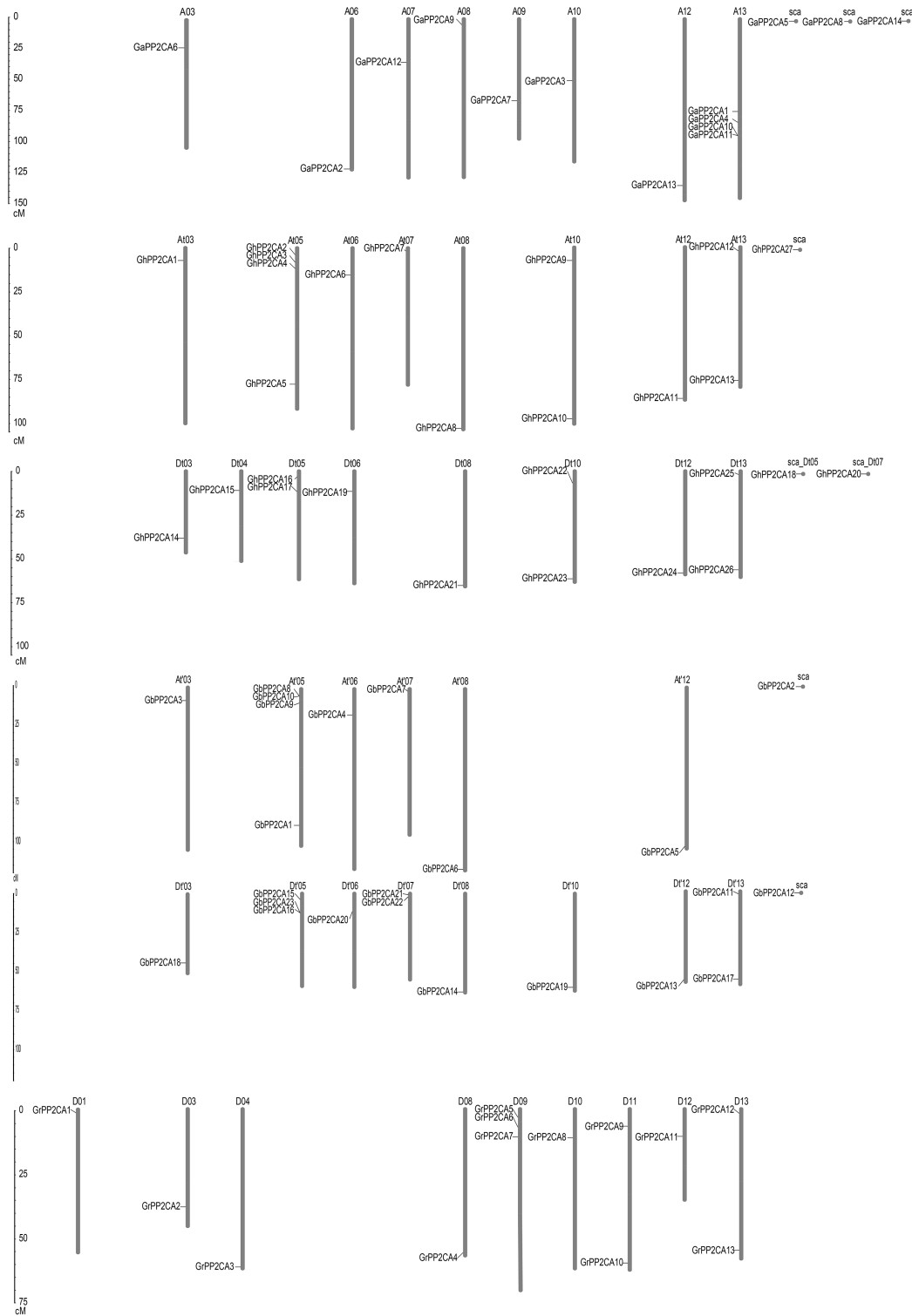

**Figure 2 Positions of *Gossypium* PP2CA genes on chromosomes.** *GaPP2CAs*, *GrPP2CAs*, *GhPP2CAs*, and *GbPP2CAs* were from *G. arboreum*, *G. raimondii*, *G. hirsutum*, and *G. barbadense,* respectively.

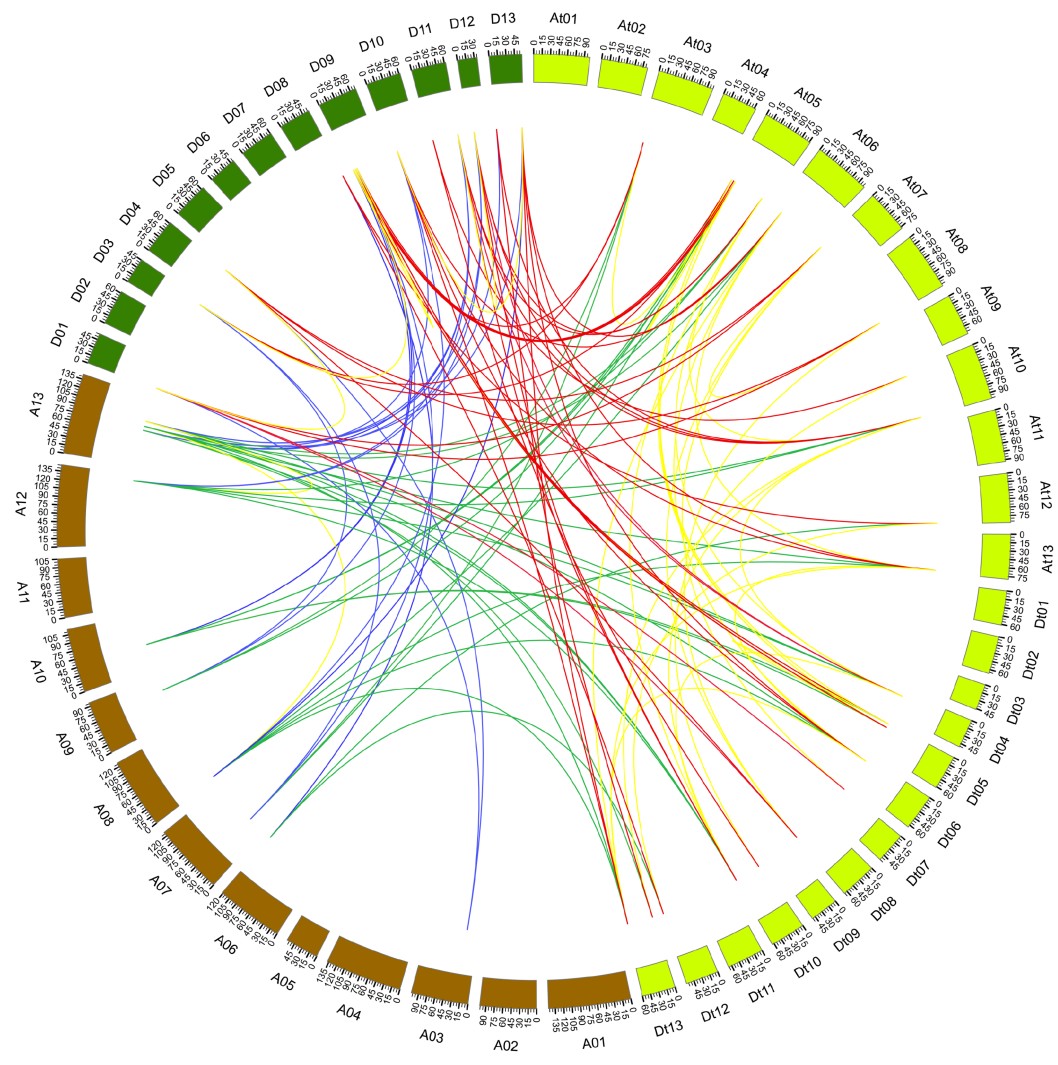

**Figure 3 Genome-wide synteny results of PP2CA genes from *G. arboreum*, *G. raimondii*, and *G. hirsutum*.** Green lines linked gene pairs between *G. arboreum* and *G. hirsutum*, red lines connected gene pairs between *G. raimondii* and *G. hirsutum*, blue lines bridged gene pairs between *G. arboreum* and *G. raimondii,* yellow lines joined gene pairs within individual species in *G. arboreum*, *G. raimondii*, and *G. hirsutum*.               

imply that segmental duplication plays major roles in generating *PP2CAs* during evolution in *Gossypium*.

## Analysis of *Ka*/*Ks* values of *PP2CAs*

To further understand the evolution processes among *Gossypium* PP2CAs, the effects of selection on duplication of *PP2CA* genes were determined. The *Ka* and *Ks* substitutions, and *Ka*/*Ks* values were calculated for the homologous gene pairs among *GaPP2CAs*, *GrPP2CAs*, and *GhPP2CAs*. The mean values of *Ka*/*Ks* for these gene pairs between species Ga/Gh, Gr/Ga, Gr/Gh, Ga/Ga, Gr/Gr, Gh/Gh were 0.22, 0.21, 0.22, 0.21, 0.19, and 0.21, respectively. All of them were less than 1, indicating that the formation of these genes were mainly under purifying selection during evolution. The *Ka*/*Ks* ratios for the two gene

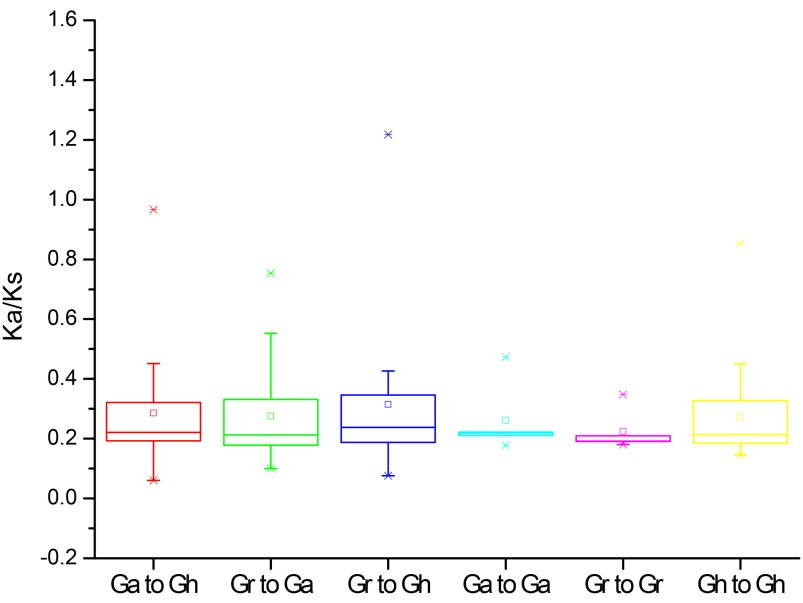

**Figure 4** The *Ka/Ks* values of the homologous PP2CA gene pairs in *Gossypium*. (Ga), (Gr), and (Gh) represented *G. arboreum*, *G. raimondii*, and *G. hirsutum*, respectively.

pairs *GrPP2CA11/GhPP2CA15* and *GrPP2CA3/GhPP2CA21* were higher than 1, suggesting that the two gene pairs were generated under positive selection and the selection likely has effects to change these genes during evolution (Fig. 4).

## Phylogenetic analysis of PP2CAs in *Gossypium* and other plants

We constructed a phylogenetic tree of PP2CA proteins in *G. arboreum*, *G. raimondii*, *G. hirsutum*, *G. barbadense*, *A. thaliana*, *T. cacao*, *R. communis*, *P. trichocarpa*, *G. max*, *B. distachyon*, and *O. sativa* using the ML method, and analyzed the evolutionary relationships of these PP2CAs. It was found that the PP2CAs included both dicotyledonous and monocotyledonous members (Fig. 5). This suggests that these PP2CAs formed before the divergence of eudicots and monocots and are in general highly conserved. Indeed, the PP2CAs from the eudicots *Gossypium*, cacao, poplar, castor, soybean and *Arabidopsis* clustered more closely, and those of the monocots rice and *distachyon* clustered together. Moreover, many PP2CAs from *Gossypium* clustered more closely with those from cacao than from poplar, castor, soybean and *Arabidopsis* (Fig. 5), indicating that PP2CAs of *Gossypium* had closer relationship with those of cacao than those of other plants. As expected, PP2CAs in the four *Gossypium* species always clustered together, in line with their homologous evolutionary relationships (Fig. 5).

## Expression patterns of *GhPP2CA* genes in different tissues

The transcript abundances of 27 *GhPP2CAs* in various tissues were measured by qRT-PCR to determine the putative functions of the PP2CAs in cotton. The results showed that all of the *GhPP2CAs* were highly expressed in flowers. *GhPP2CA11* and *GhPP2CA27* were also preferentially expressed in roots. Moreover, the transcriptional levels of *GhPP2CA3*,

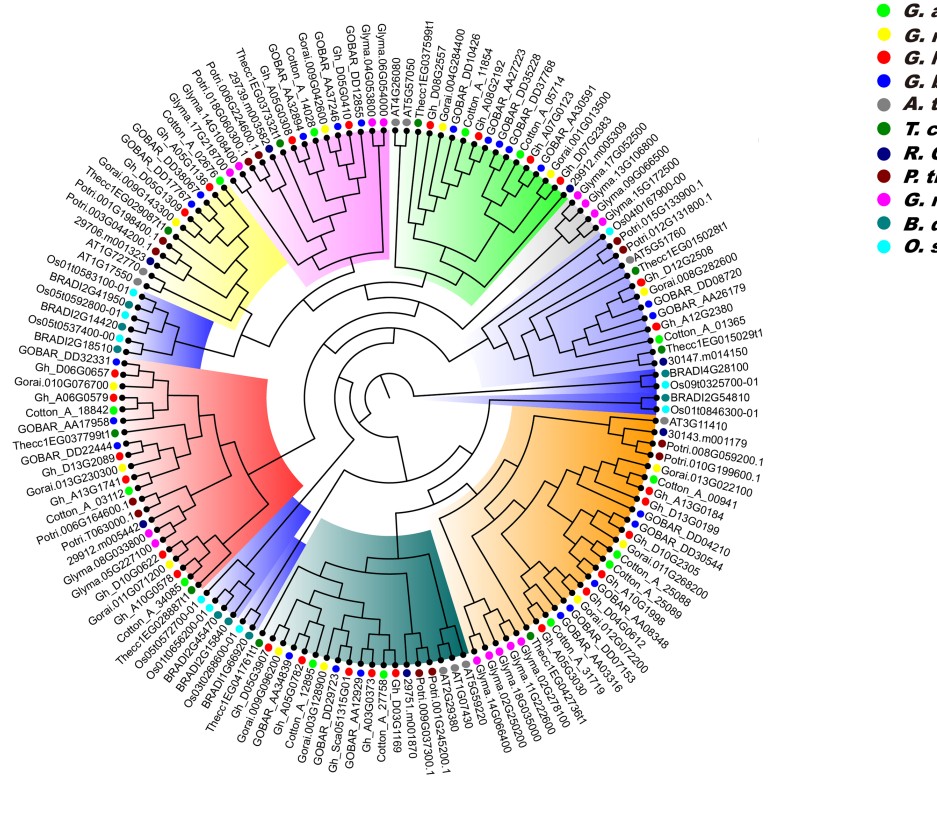

2.0

**Figure 5** **Phylogenetic tree of PP2CAs in *Gossypium* and other plants.** The multiple alignment was performed by MUSCLE program. IQTREE was used to create the maximum likelihood with Dayhoff model. The phylogenetic relationship of PP2CAs in *G. arboreum, G. raimondii, G. hirsutum, G. barbadense, A. thaliana, T. cacao, R. communis, P. trichocarpa, G. max, B. distachyon*, and *O. sativa* were analyzed.                               

*11*, *13*, *27* were high in fibers. The transcripts of *GhPP2CA4*, *16*, *22* were abundant in stems. These results imply that most cotton PP2CA members may function in reproductive development, and some PP2CAs also play roles in some specific tissues like roots, fibers, and stems (Fig. 6).

## Transcriptional changes of *GhPP2CAs* in responses to ABA and osmotic stress

To gain insight into the roles of *GhPP2CAs* in ABA signaling, transcriptional abundances of *GhPP2CAs* in roots were detected after treatments with 100 µM ABA or 10% PEG6000 for indicated periods of time. We observed that the transcriptional levels of some *GhPP2CA* genes such as *GhPP2CA5*, *11*, *18*, *20*, *25*, *27* continually increased with the extension of ABA treatment time. In contrast, the expression levels of some members including *GhPP2CA2–4*, *8*, *10*, *24* had decreasing trends. The expression levels of some genes were decreased at 3 or 6 h but increased at 12 or 24 h. These genes included

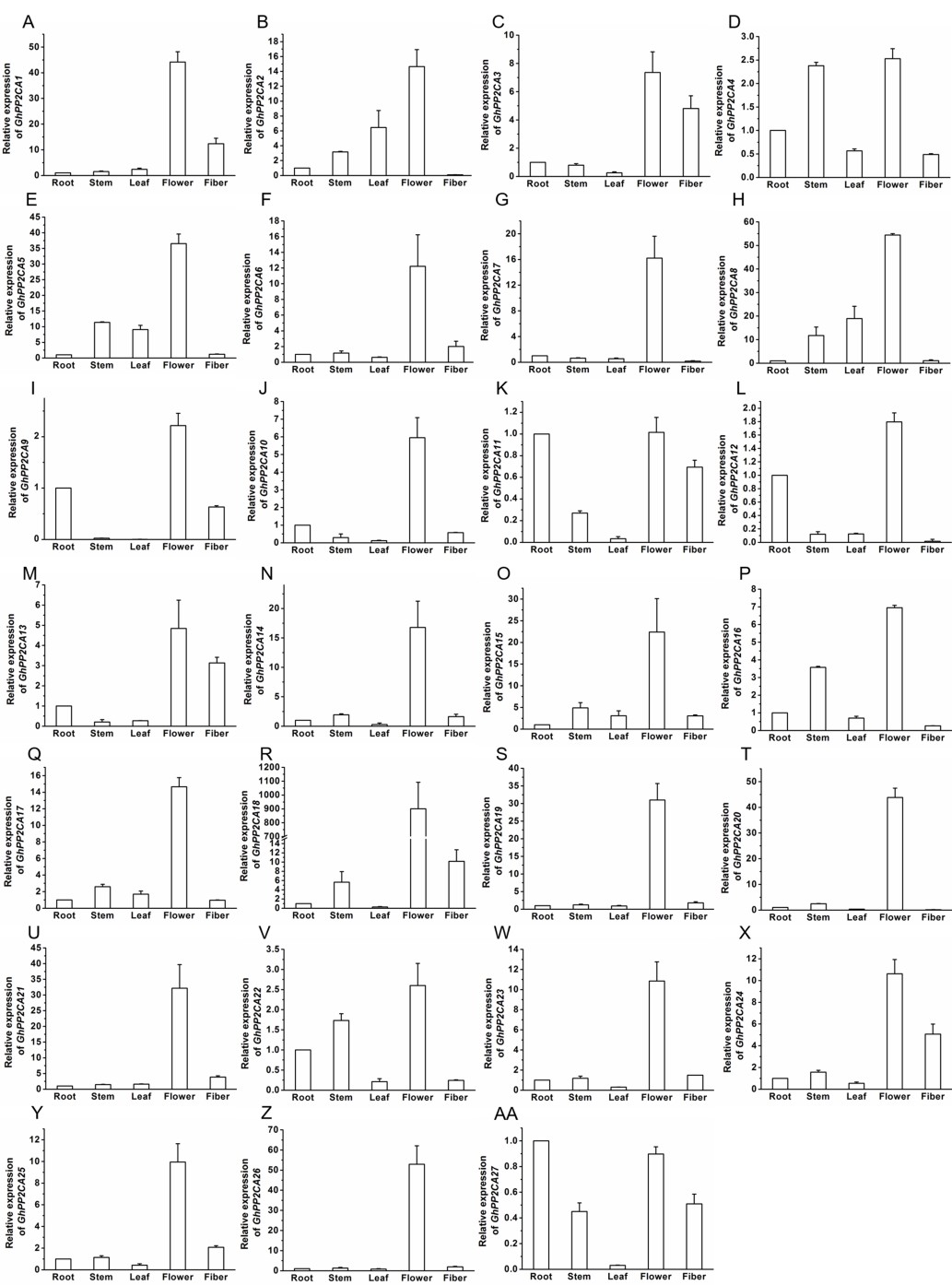

**Figure 6 Transcription levels of *GhPP2CAs* in cotton tissues.** The genes preferentially expressed in flowers (A–Z) and fibers (AA). Gene *GhUBQ7* was used as the internal control. The expression level of the gene in roots was set as 1. The data were mean ± SE. Statistical analyses were carried out by student's *t*-test to determine the differences in gene abundances between roots and other tissues.

*GhPP2CA1, 16, 17, 19, 26.* The expression levels of several genes increased at 3 or 6 h but decreased at 12 or 24 h. These genes were *GhPP2CA6, 7, 9, 12–15, 21–23* (Fig. 7). Treatment of cotton seedlings with PEG6000 also altered the expression of most

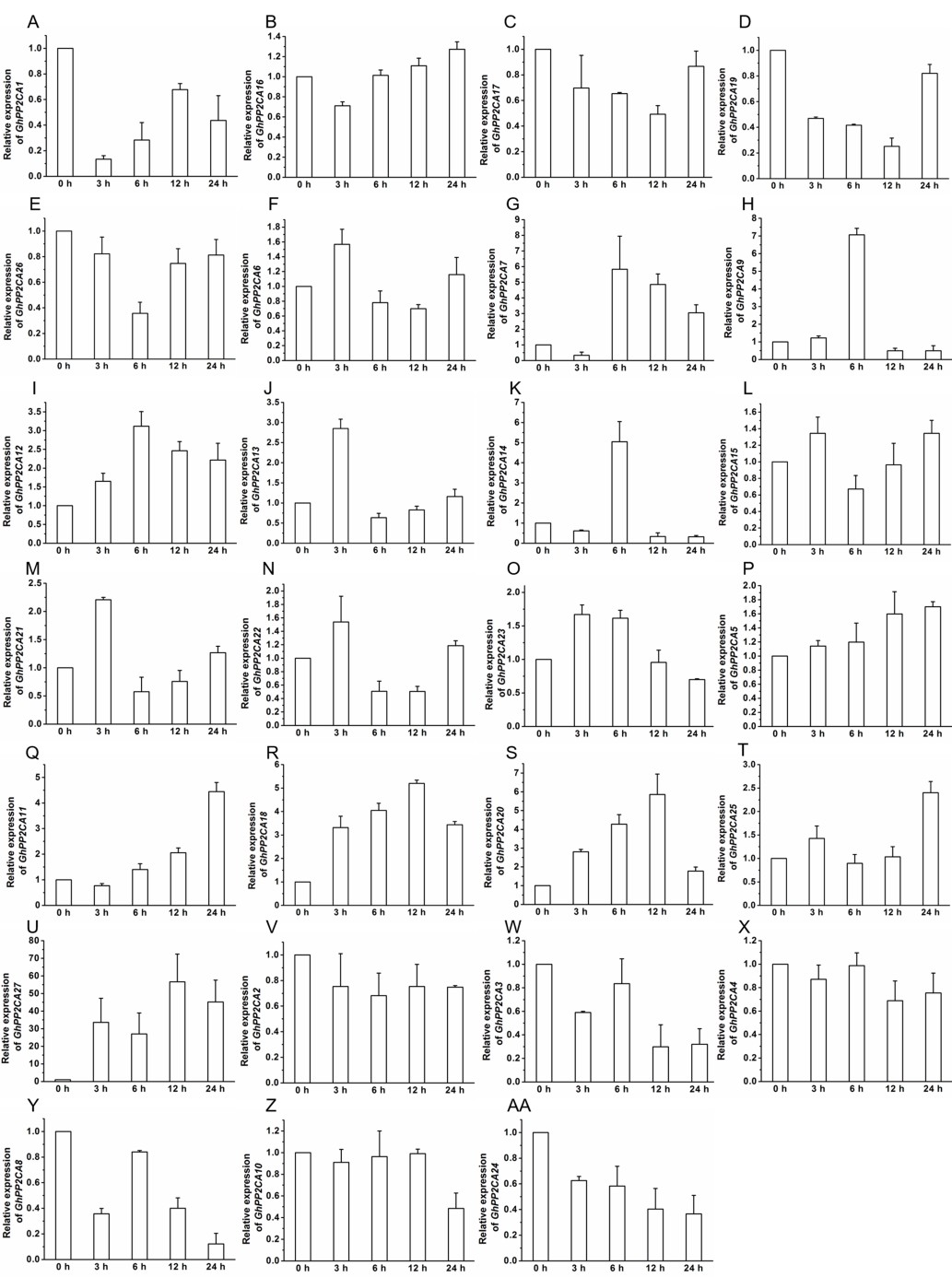

**Figure 7 Expression of *GhPP2CA* genes in response to ABA.** The relative expression of *GhPP2CAs* was monitored after treatments with 100 µM ABA for indicated periods of time. The expression levels of the genes were prominently reduced at 3 or 6 h but elevated at 12 or 24 h (A–E), and were increased at 3 or 6 h but decreased at 12 or 24 h (F–O). The transcription abundances of the genes were generally enhanced (P–U), and diminished (V–AA), respectively, with the prolongation of treatment time. Cotton gene *GhUBQ7* was applied as the internal control. The gene expression value at 0 h was set as 1. The values were mean ± SE. Statistical analyses were carried out by student's *t*-test to determine the differences in gene abundances between 0 h and other treatment times.

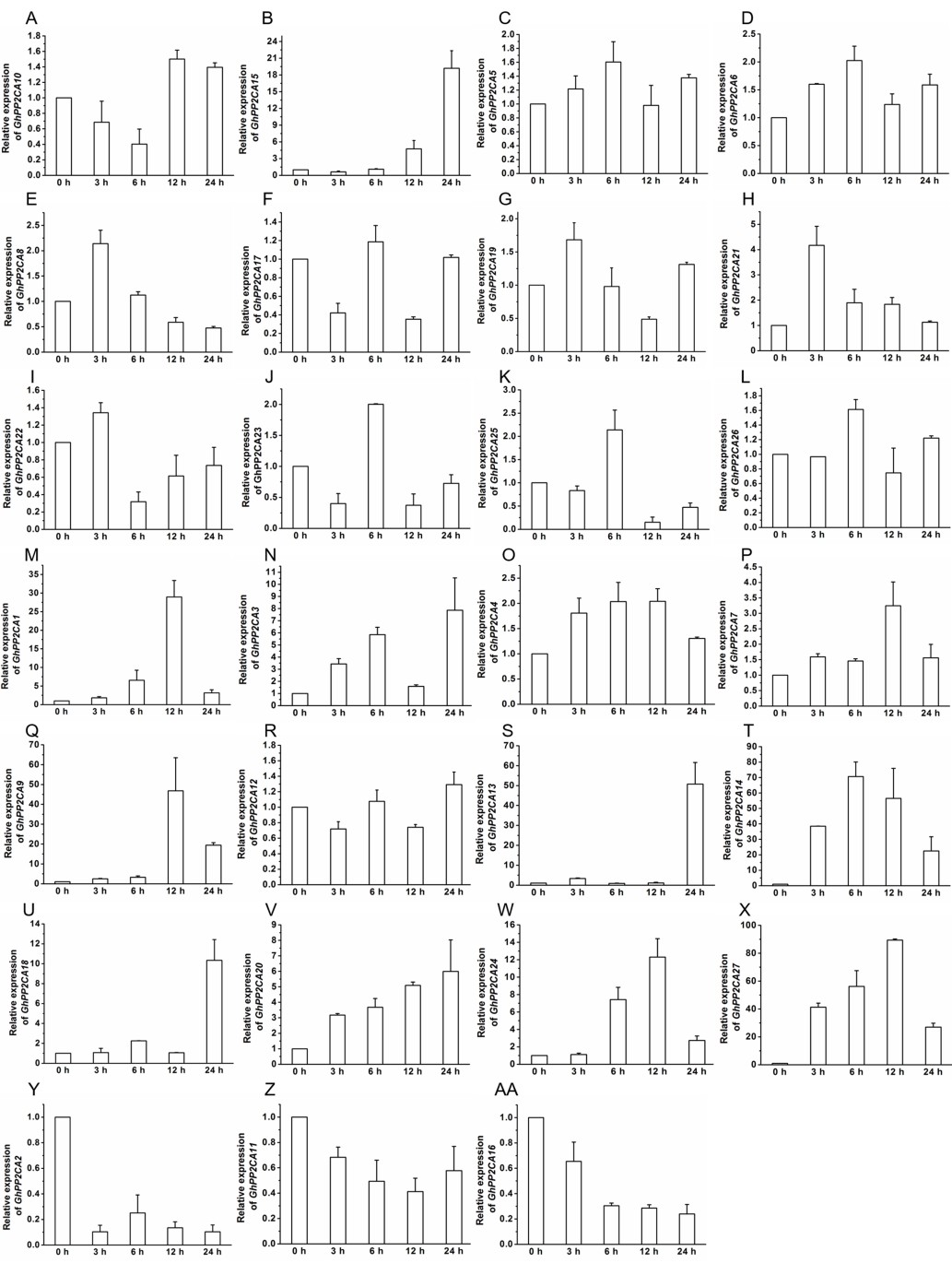

**Figure 8 Expression of *GhPP2CA* genes in response to osmotic stress.** The relative expression of *GhPP2CAs* was determined after exposure upon 10% PEG6000 for indicated time. The transcriptional levels of the genes were clearly decreased at 3 or 6 h but increased at 12 or 24 h (A and B), and rose at 3 or 6 h but dropped down at 12 or 24 h (C–L), the expression of the genes in (M–X) was elevated, and that in (Y–AA) was reduced with the extension of treatment time. *GhUBQ7* was used as the internal control. The gene expression value at 0 h was set as 1. The values were mean ± SE. Statistical analyses were carried out by student's *t*-test to determine the differences in gene abundances between 0 h and other treatment times.

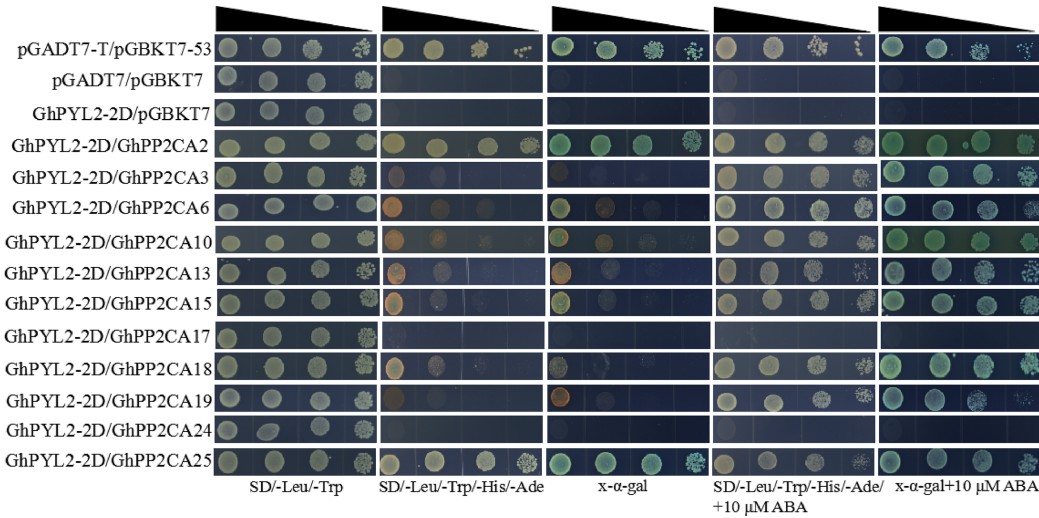

**Figure 9** **Analysis of interactions between GhPP2CAs and GhPYL2-2D by yeast-two hybrid method.** Narrowing triangles indicate the reduced cell densities in the dilution series from left to right.

GhPP2CA genes (Fig. 8). The majority of *GhPP2CAs* were upregulated after treatments with PEG for a short time period and downregulated afterward. For example, the transcriptional levels of *GhPP2CA8* and *GhPP2CA21* were prominently enhanced at 3 h, and then reduced at 6, 12, and 24 h, while those of *GhPP2CA5, 6, 17, 23, 25, 26* were pronouncedly increased at 6 h and decreased at 12 and 24 h. By contrast, the expression of some genes was significantly elevated at 12 or 24 h post PEG treatment. These genes included *GhPP2CA1, 3, 4, 7, 9, 12–14, 18, 20, 24, 27*. The overall trend of expression of *GhPP2CA10* and *GhPP2CA15* was increased while that of *GhPP2CA2, 11, 16* was decreased under osmotic stress (Fig. 8). Together, these data suggest that *GhPP2CAs* exhibit diverse expression patterns in responses to ABA and osmotic stress.

## Many GhPP2CAs interact with GhPYL2-2D, GhPYL6-2A, and GhPYL9-2A

Clade A type 2C protein phosphatases have been documented to interact with ABA receptor PYLs in ABA signal pathway (*Ma et al., 2009*; *Park et al., 2009*). Accordingly, we investigated the interactions between GhPP2CAs and GhYPLs in the absence or presence of ABA by yeast-two hybrid method. A total of 11 *GhPP2CAs* were cloned, and three *GhPYLs GhPYL2-2D* (Gh_D08G2587), *GhPYL6-2A* (Gh_A06G1418), and *GhPYL9-2A* (Gh_A11G0870) were randomly selected and cloned. These genes were fused into yeast vectors, and yeast-two hybrid experiments were performed. In the absence of ABA, GhPP2CA2, and GhPP2CA25, respectively, interacted with GhPYL2-2D while multiple GhPP2CAs like GhPP2CA2, 3, 6, 10, 13, 15, 18, 19, 25 individually interplayed with GhPYL2-2D in the presence of ABA (Fig. 9). In contrast, several GhPP2CAs could, respectively, interact with GhPYL6-2A or GhPYL9-2A either with or without ABA. These GhPP2CAs included GhPP2CA2, 6, 10, 13, 15, 18, 19, 24, 25. Besides, GhPP2CA3 interact with GhPYL9-2A but not with GhPYL6-2A either in the presence or absence

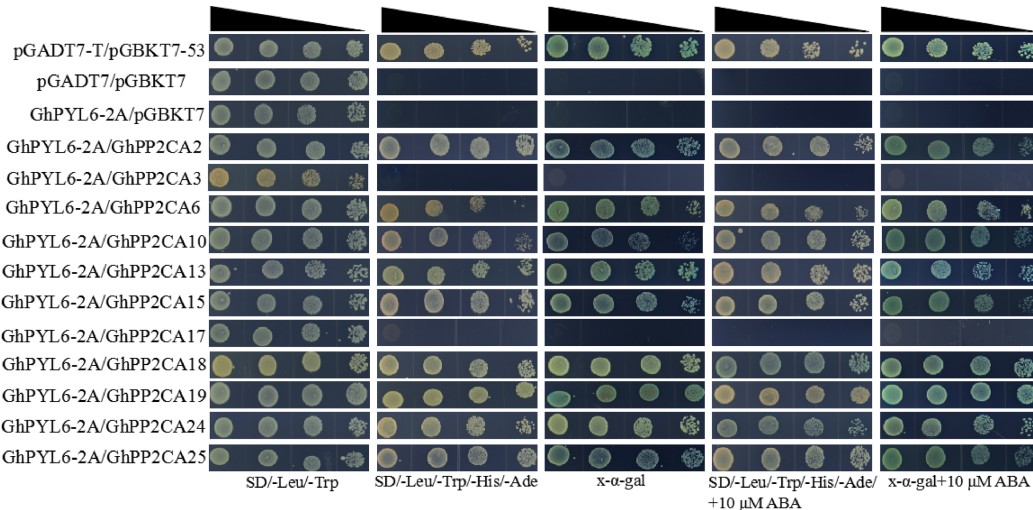

**Figure 10 Assay of interactions between GhPP2CAs and GhPYL6-2A using yeast-two hybrid technology.** Narrowing triangles show the reduced cell densities in the dilution series from left to right.

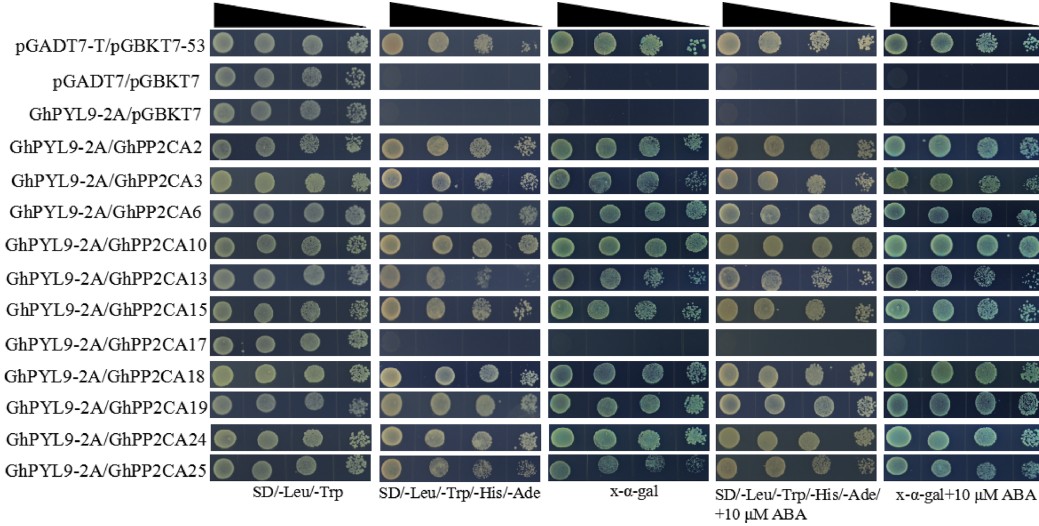

**Figure 11 Determination of interactions between GhPP2CAs and GhPYL9-2A by yeast-two hybrid method.** Narrowing triangles reveal the reduced cell densities in the dilution series from left to right.

of ABA (Figs.10 and 11). These results imply that GhPP2CAs differentially interact with GhPYLs in responding to ABA in cotton.

## DISCUSSION

Clade A type 2C protein phosphatases are central components of ABA signal transduction pathway, and negatively control ABA and stress responses in plants (*Fuchs et al., 2013*; *Singh et al., 2015*). They have been identified in several plants including *Arabidopsis*, rice, maize and *B. distachyon* in recent years (*Xue et al., 2008*; *Wei & Pan, 2014*; *Cao et al., 2016*). However, phylogenesis and putative functions of PP2CAs in *Gossypium* remain

elusive. In the present study, 14, 13, 27, and 23 PP2CA genes were characterized in genomes of *G. arboreum*, *G. raimondii*, *G. hirsutum*, and *G. barbadense*, respectively (Table 1). Compared to the number of PP2CAs in *Arabidopsis* (9), rice (10), maize (16), and *B. distachyon* (8), that in *G. hirsutum* and *G. barbadense* was great (*Xue et al., 2008*; *Wei & Pan, 2014*; *Cao et al., 2016*). This suggests that more complex and elaborate ABA signaling mechanisms modulated by PP2CAs may exist in the upland and island cotton species. Conceivably, the high number of PP2CAs of the two species is related to their tetraploid nature. The two plants retain most PP2CA homologs of both *G. arboreum* and *G. raimondii* but not a copy of either progenitor during evolution. This may be due to long-term human selection with these two tetraploid cotton species for higher yields, growth in hotter and drier regions, day neutral flowering, and adaptation to agronomic areas far outside their original habitats. These altered characteristics may be associated with more PP2CA proteins and complex ABA signal mechanisms in the cultivated cotton plants than in wild plants and in the greater opportunities to accumulate sequences, amenable to mutation and selection, in a tetraploid genome than in a diploid genome.

We noticed that 27 GhPP2CAs and 23 GbPP2CAs individually had their corresponding orthologs in *G. arboreum* or *G. raimondii* (Fig. 1), indicating that those PP2CAs in *G. hirsutum* and *G. barbadense* are ancestrally related to those PP2CAs in the two diploid species. Additionally, no orthologous genes of *GaPP2CA1, 3, 4, 14, GrPP2CA6*, and *GrPP2CA9* were observed in *G. barbadense* (Fig. 1). This hints that these genes are possibly lost, or these genes arose after the tetraploid species appeared and separated from the diploid species during the evolutionary processes.

The structures and the numbers of introns and exons in *PP2CAs* were similar among the four *Gossypium* species as well as *Arabidopsis*, rice, maize and *B. distachyon* (*Xue et al., 2008*; *Wei & Pan, 2014*; *Cao et al., 2016*), suggesting that the *PP2CAs* undergo conserved evolutionary processes even after the divergence of monocotyledons and dicotyledons. Colinearity results showed that 136 homologous gene pairs existed among *GaPP2CAs*, *GrPP2CAs*, and *GhPP2CAs* (Fig. 3), indicating that PP2CA genes expand primarily through segmental duplication of DNA. Segmental duplicates may be more often maintained through subsequent gene subfunctionalization compared to tandem duplicates (*Lynch & Conery, 2000*). Accordingly, these PP2CAs probably had diverse functions in *Gossypium*. Moreover, in agreement with our results, *Arabidopsis* phosphatase family genes also showed segmental duplication (*Cannon et al., 2004*), suggesting the evolutionary mechanism of PP2CAs may be conserved in plants. The mean value of $Ka/Ks$ for a majority of *PP2CA* homologous gene pairs was about 0.2, significantly less than 1 (Fig. 4). This hints that most mutations occurred in the genomic sequences of *PP2CAs* in *G. arboreum*, *G. raimondii*, and *G. hirsutum* during evolution were detrimental for plant survival, or the mutations-caused traits were not required for man. Thus, these mutated genes were gradually eliminated, and only those we found were kept during the long-time selection.

Phylogenetic results showed that the PP2CA members from monocotyledonous plants clustered together, and similar results occurred in dicotyledonous PP2CAs (Fig. 5).

This suggests that great changes in DNA sequences of the *PP2CAs* have taken place after isolation of monocotyledons and dicotyledons although these genes shared a common ancestor. PP2CAs in *Gossypium* always clustered together with those in *T. cacao* rather than with those in *A. thaliana*, *R. communis*, *P. trichocarpa*, *G. max*, *B. distachyon*, and *O. sativa* (Fig. 5), pointing to the closer evolutionary relationship of *Gossypium* with *T. cacao*. That is, most of homologous PP2CA members in *Gossypium* and *cacao* were generated before separation of the two genera from the common ancestor. The common PP2CAs across all plants show that they still have core functions essential to basic plant survival and functions. Differences in PP2CAs that follow differentiation of different genera and even species show that they are still diverse and can accommodate functions specific to the survival of species and even in response to selection by man. "Housekeeping" PP2CAs could probably be subtracted from the picture to illuminate the more unique ones to better understand functions of individual PP2CAs and their roles in specific species, traits, or even agronomic performance of specific cultivars.

Transcript abundance analysis indicated that the majority of the *GhPP2CAs* was predominantly expressed in flowers (Fig. 6), suggesting that GhPP2CA-mediated ABA signaling may be of great importance in flower development of cotton. High expression of *GhPP2CAs* in flowers was likely due to the importance of timing flowering to environmental conditions of native *Gossypium* plants. Because evolution of some species is tied to long-term human selection of cotton plants with high yields of fibers and good adaptations to hot and dry growth conditions. Cotton yields are closely associated with flowering in agronomic conditions created by man, which often are much different from the natural habitats of wild or ancestral *Gossypium* species (e.g., cultivation only in summers of temperate–tropical latitudes instead of perennial growth in tropical latitudes closer to the equator). Drought and hot stresses should limit flower development. PP2CAs are negative regulators of the adverse stresses; and therefore, may facilitate flowering of cotton in these newer environments. The expression of most *GhPP2CAs* was upregulated after treatment with ABA or PEG6000 (Figs. 7 and 8), in good agreement with the results from *AtPP2CAs*, *OsPP2CAs*, and *BdPP2CAs* (Xue et al., 2008; Cao et al., 2016). These findings imply that PP2CAs are essential for plant response to ABA and osmotic stress.

The interactions between 11 GhPP2CAs and three GhPYLs were examined. The results revealed that most GhPP2CAs can individually interact with the three GhPYLs in the absence or presence of ABA (Figs. 9–11). GhPYLs are homologs of AtPYLs and some GhPYLs have been suggested to be functional ABA receptors (Liang et al., 2017; Zhang et al., 2017b). These data indicate that a large number of GhPP2CAs may play roles via interactions with GhPYLs in ABA-dependent or ABA-independent manner in cotton. The detailed mechanisms of GhPP2CAs in ABA signaling will be further explored in the future.

## CONCLUSIONS

In total, 14, 13, 27, and 23 PP2CA genes were characterized from *G. arboreum*, *G. raimondii*, *G. hirsutum*, and *G. barbadense*, respectively. These genes shared high similarity in chromosomal locations, structures, and phylogeny among the species. Most of them might be under purifying selection during evolution. Moreover, *PP2CAs*

displayed specific expression patterns in tissues and diverse expression profiles in response to ABA and osmotic stress in *G. hirsutum*. Yeast-two hybrid experiments indicated that most GhPP2CAs interacted with GhPYL2-2D, GhPYL6-2A, and GhPYL9-2A with or without ABA. These findings provide essential information for in-depth investigations of the functions of PP2CAs in *Gossypium* in the future.

### Funding
This work was supported by Foundation of Program for Young Backbone Teachers in Universities of Henan Province (2016GGJS-024), the Scientific Research and Innovation Team in Henan University of Animal Husbandry and Economy (2018KYTD18), and the "111" Project. The funders had no role in study design, data collection and analysis, decision to publish, or preparation of the manuscript.

### Grant Disclosures
The following grant information was disclosed by the authors:
Foundation of Program for young backbone teachers in universities of Henan Province: 2016GGJS-024.
Scientific Research and Innovation Team in Henan University of Animal Husbandry and Economy: 2018KYTD18.
"111" Project.

### Competing Interests
The authors declare that they have no competing interests.

### Author Contributions
- Tingting Lu performed the experiments, analyzed the data, prepared figures and/or tables, approved the final draft.
- Gaofeng Zhang performed the experiments, prepared figures and/or tables.
- Yibin Wang analyzed the data.
- Shibin He analyzed the data, authored or reviewed drafts of the paper.
- Lirong Sun conceived and designed the experiments.
- Fushun Hao conceived and designed the experiments, contributed reagents/materials/analysis tools, authored or reviewed drafts of the paper, approved the final draft.

### Data Availability
Gene primers used for quantitative real-time RT-PCR experiments and yeast-two hybrid experiments are available in Tables S1 and S2. PP2C members in cotton are available in Table S3.

### Supplemental Information
Supplemental information for this article can be found online at http://dx.doi.org/10.7717/peerj.7105#supplemental-information.

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
