# Peer review of "Genome-wide characterization and expression analysis of PP2CA family members in response to ABA and osmotic stress in Gossypium"

_PeerJ, doi:10.7717/peerj.7105_

## Round 0.1 · original submission · Minor Revisions

I have received two detailed reviews and am prepared to move forward with a decision of minor revision. Before I give any details I would like to offer an apology; I wasn't feeling well over the holidays and, frankly, I got behind. I feel horrible that I have delayed this decision and promise to do better when you resubmit.

I feel that all of the comments from the reviewers are well taken. I have one additional issue that I would like you to address - the phylogenetic analysis. In general, I dislike the use of the neighbor joining (NJ) method. If it is used I feel it is critical to describe the distance used, the description in the paper does not provide this information. You state:

"The amino acid sequences of PP2CAs were aligned, and a phylogenetic tree of the PP2CAs was generated following the neighbor joining method (Neighbor-Joining, NJ). One thousand bootstrap trials with the Clustal W tool (Larkin et al., 2007) and the MEGA 5.0 software (http://www.megasoftware.net/) were used."

Was NJ done with ClustalW? Or was ClustalW invoked within MEGA? This is very unclear.

I would like you to repeat the phylogenetic analysis using an alignment program that has been shown to perform better than ClustalW (either MUSCLE or MAFFT) and generate a maximum likelihood phylogeny. There are web servers for both MUSCLE (https://www.ebi.ac.uk/Tools/msa/muscle) and MAFFT (https://mafft.cbrc.jp/alignment/server/). I have a slight personal preference for MUSCLE but both programs perform well in benchmarks so the choice is ultimately yours. However, you should cite the papers describing the method, not the web servers (i.e., Katoh, Rozewicki, Yamada 2017 (Briefings in Bioinformatics, in press) MAFFT online service: multiple sequence alignment, interactive sequence choice and visualization. and Katoh, Toh 2008 (Briefings in Bioinformatics 9:286-298) Recent developments in the MAFFT multiple sequence alignment program. for MAFFT --or-- Edgar, R.C. (2004) MUSCLE: multiple sequence alignment with high accuracy and high throughput. Nucleic Acids Res. 32(5):1792-1797 for MUSCLE).

For the actual phylogenetic analysis I would strongly suggest the use of IQ-TREE (http://www.iqtree.org/). There is a link to an IQ-TREE web server at the bottom of the IQ-TREE web site. Use AUTO model selection (which will test the full set of protein models implemented in IQ-TREE and choose the best-fitting model) and use the ultrafast bootstrap to examine support. Again, please cite the papers describing the IQ-TREE program, web server, the paper describing the ultrafast bootstrap, and the paper describing whichever model ends up being the best model.

I think all other areas of the manuscript are well covered by the reviews. I look forward to seeing a revised version of this interesting paper!

Ed

·

Basic reporting

Abstract
It is not clear what GhPYL2-2D, etc. are? Are they Genebank sequences of PP2CA sequences or another gene, localized to subgenomes D08, A06, and A11? The introduction says that PP2CAs are co-receptors, interacting with PYL and controlling ABA signaling. I would somehow mention this in a sentence in the abstract so the audience understands the relationship and further understands the reason for the yeast two-way hybrid experiments.

Line 38 were identified the genomes of these plants, respectively. Analysis results showed that most
CHANGE ‘identified the genomes’ to were identified in the genomic sequences

Line 42 PP2CAs were under purifying selection in Gossypium during evolution. Moreover, majority of the PP2CAs were expressed specifically in diverse tissues, and highly expressed in flowers in G.
CHANGE ‘Morever, majority’ to The majority

Line 45 osmotic stress. Besides, yeast-two hybrid assays revealed that many GhPP2CAs could interact
CHANGE ‘Besides’ yeast-two hybrid assays revealed that many GhPP2CAs could interact’ to Yeast-two hybrid assays revealed that many GhPP2CAs are capable of interaction with

Line 79
2013; Singh et al., 2015). PP2CAs are functional redundant, and their expression is upregulated
CHANGE ‘functional’ to functionally

Line 119 default setting (E-value<e-10). The putative Gossypium PP2Cs were then characterized using the
DEFINE putative. What criteria lead to the selection of sequences and how many sequences for the next step? The output of BLAST can be extensive so what is the cutoff?

Line 123 The online tool ExPaSy (http://web.expasy.org/protparam/) was applied to analyze the
Line 124 properties of PP2CAs in Gossypium. The subcellular localizations of Gossypium PP2CAs were predicted using the program in the website (http://www.csbio.sjtu.edu.cn/bioinf/Cell-PLoc/). The
Line 126 structures of the PP2CAs were characterized by the GSDS (http://gsds.cbi.pku.edu.cn/). The
COMMENT That is a hefty lineup of methods. Add some more information to each method as to the purpose, any other settings used, and cutoff values or quality control measure to continue to the next method.

Line 143 We downloaded the PP2CA databases for Arabidopsis thaliana (http://www.arabidopsis.org/),
CHANGE ‘We downloaded the PP2CA databases for’ to The PP2CA databases were downloaded for

Line 155
collected from TM-1 cotton plants grown in soil for 21 d. Flowers were got 1 d post anthesis, and
CHANGE ‘Flowers were got’ to Flowers were collected
DESCRIBE source of TM-1 because specific clones give different results. I know the US Cotton Germplasm Collection offers one as SA-2269, PI 607172, but individual labs may have kept their own clones.

Line 158 MS medium in a growth chamber (day/night temperature cycle of 28°C/26°C, 14 h light/10 h
ADD citation for MS medium

Line 208 3A’, GbHAI3-3D, GbHAI3-3A, GbABI1-3D) had not orthologs in G. arboretum and G.
CHANGE ‘not’ to no

233 GhHAI2D, GhABI2-1D, GbAHG3-2A and GbAHG3-1D were located on scaffolds rather than
CLARIFY the meaning of ‘scaffold’. Is it unassigned to a chromosome?

263 under purifying selection during evolution. The Ka/Ks ratios for the two gene pairs GrAHG3-
KEEP the italics consistent, because Ka/Ks was italicized in line 259, but not in lines 260 and 263.

Line 271 members. This suggests that these PP2CA formed before the divergence of eudicots and monocots.
COMMENT. Figure 5 should be highlighted for the fact that, as expected, Gossypium sequences clustered together before another genus comes close. Were the Arabidopsis sequences in this figure also in S1- S4? Those that clustered with a Theobroma sequence, are they on the same locus in Gossypium, because it suggests stability across genera and should be within the genus too. If not, then they duplicate by rearrangements or transposable elements, etc. The whole block of just G. barbadense sequences that first cluster with Oryza before another Gossypium species is very curious to me as it seems specific to that species.

Line 312 GhPP2CAs exhibit diverse expression patterns in responses to ABA and osmotic stress.
QUESTION, you mentioned that there is interaction between PP2CAs, so is there a cascade of interactions of those active at 3h, followed by 6h, followed by 12h and 24h? It is logical that after 3h, those activated or upregulated will trigger those you see active or upregulated at 6h, and so on.

Line 338 Compared to the number of PP2CAs in Arabidopsis
Line 339 (10), rice (10), maize (16) and Brachypodium distachyon (8), that in G. hirsutum and G.
Line 340 barbadense was great (Xue et al., 2008; Wei and Pan, 2014; Cao et al., 2016). This suggests that more complex and elaborate ABA signaling mechanisms modulated by PP2CAs may exist in the upland and island cotton species.
SUGGESTION, emphasize the polyploid nature of cotton and how this could have allowed more duplication in sequences, while still conserving one copy of the original. Also it was adapted to the needs of humans, growing in many regions of the Americas. Cotton is often grown in hot dry regions because it benefits boll ripening and harvesting of dry, high quality fiber so evolution and selection among these sequences could help fine tune cotton to the farmers needs. Remember G. raimondii is wild and is photoperiodic in its flowering so I would expect differences here. Having grown G. raimondii in greenhouses I noticed that it easily drops its flowers so I wonder if ABA is functioning here? Flowering and development of bolls is a resource intensive process in cotton, even in hot and dry weather conditions so maybe they were selected by man for such areas, when it runs contrary to native Gossypium plants that normally flower in short daylength and cooler winter periods of the tropics, not hot, long days of semi-tropical regions.

352 and G. barbadense are directly descended from those PP2CAs in the two diploid species.
SUGGESTION reword ‘directly descended from those’ to ancestrally related to those

Line 355 Additionally, no orthologous genes of
Line 356 GaAHG3-3, GrABI1-2 and GrHAI2 were observed in G. barbadense (Fig. 1). This hints that these genes are possibly lost during evolution.
COMMENT Another possibility is that these genes arose after the tetraploid species appeared and are the result of separate evolutionary processes in these diploid species.

361 evolutionary processes even after the divergence of monocotyledon and dicotyledon.
CHANGE ‘monocotyledon and dicotyledon’ to monocotyledons and dicotyledons

371 Transcript abundance analysis indicated that majority of the GhPP2CAs predominantly
CHANGE ‘that majority of the GHPP2CAs predominantly’ to that the majority of the GhPP2CAs were predominantly

Lin 384 Totally, 14, 13, 27 and 29 PP2CA genes were characterized from G. arboretum, G. raimondii, G.
CHANGE ‘totally’ to In total


Literature references, sufficient field background/context provided.

I am impressed with the mastery of the English language. The structure is logical and it flows well. The introduction was easy to follow. I would consider finding some citations that show that ABA or protein phosphatases were implicated in cotton yield, physiology, disease resistance, etc. so it appeals to a wide audience. By itself the research is fascinating and a great contribution to basic science but some are reading it with an idea of application to cotton improvement. What follows is a listing of grammatical changes and comments that I accumulated while reading the article.

Line 101 The yield and quality of cotton are adversely affected by many abiotic
Line 102 stresses such as drought and high salinity, which are governed by ABA signaling. Therefore, it is
ADD references to this statement that show the current state of knowledge of ABA in cotton.

Line 118 (http://database.chgc.sh.cn/cotton/index.html), respectively, using the BLAST program with
ADD the reference for BLAST

Ohta et al., 2003 should go after Park et al., 2009 in your citation section because they are all arranged alphabetically by main author’s last name. Otherwise it appears that the citations in the text are all spelled out in the citation section.


Professional article structure, figs, tables. Raw data shared.

The figures, and tables were essential to the article.

COMMENTS, in Figure 6. You display resolution in the regulation of different PP2CAs but what is the relevance or what could be happening? The differences in flower and fiber should be relevant for cotton breeders so they can separate fiber quality genes (not shutting down fiber genes will allow fibers to mature and elongate fully) from those just for yield (flowers that remain would increase total yield)? G. raimondii does not produce relevant fiber so those sequences could be further subtracted so breeders just narrow it down to those affecting fiber quality? Are they separated by chromosomes and easily selected by a breeder or scientist or are they linked? I am going to speculate here, osmotic stress normally shuts down flowering but in nature a dry period is sometimes best for flowering and perhaps Gossypium shares this feature. The pollen is sensitive to moisture and is heavy, requiring insect pollination, further indications that even though moisture is limiting and that normally restricts vegetative growth (possible releases of ABA), it must counter it with PP2CAs in order to continue flowering? Short daylengths in dry winter months are common in the tropics and normally when native Gossypium species flower so there might be a connection or one that can be investigate with the new data and tools you developed?

Table 1.
Size(aa and Mass(kDa
) )
Should be reformatted to
Size Mass
(aa) (kDa)

COMMENT, what is the meaning of ‘? For example what is the difference between GaAHG3-2 and GaAHG3-2’?

Figures 6, 7, and 8
QUESTION, how did you calculate the Standard error? I saw mention of experiments replicated three times, but how were they replicated? What are the actual data and where is mention of an ANOVA of the data?

All appropriate raw data has been made available in accordance with our Data Sharing policy.

I am not as familiar with this journal or how data is presented. Is there access to the sequences you found, because it could be difficult for a research to get the exact same sequences, even if they duplicated every thing exactly because of the size of the genomes, how one selects output and what order they use it, etc.

COMMENT, figures S1-S4, I would expect every sequence of Gossypium to cluster with one another Gossypium sequence before even coming close to a sequence in Arabidopsis. So how did some Gossypium sequences cluster with an Arabidopsis sequence before clustering with another Gossypium sequence? Maybe I am noticing the obvious but it would be good to have an explanation accompany the nice figures. It would suggest to me that certain enzymes in certain places on certain chromosomes may predate the evolutionary separation of these two families and therefore there is a grandfather sequence that ended up in both families. Evolution protected them to the extent that they resemble that sequence in other families more strongly than duplicated and descendant sequences within the same species. When you have a close association of a putative sequence in two families, but with one that is highly annotated as in Arabidopsis, it makes a compelling case for assigning genetic function.


Self-contained with relevant results to hypotheses.

I appreciate the whole package of the research as it started with genomic sequences and ended up with putative genes. It should be made clearer to the reader the purposes and details of the extensive methodology. The reader can benefit from an explanation of why each step is necessary and how they feed into each other. Then the hypothesis structure becomes more relevant. When each step continues to suggest the function as PP2CAs then your conclusions are stronger and keeps narrowing down the role of each sequence in the biochemistry of the plant. It allowed you to do the PCR primer reactions, two-way hybrid experiments, etc. all from starting just with 4 genomic sequences.

It would appear that the hypotheses are something like, phosphatases assigned by function and genetic sequences in well studied plants like Arabidopsis will confer gene function to similar genetic sequences in Gossypium. Genetic sequences in four species with published genomes that show homology to known phosphatase genes in other families can assign these broad functions to these new sequences in Gossypium. A wide assortment of research tools are used to find the closest homologies of sequences to those known to function in phosphatases and ABA, because they have strong effects in cotton physiology (references specific to cotton are needed, you have one related to potassium, which is interesting as potassium is often consumed the highest during boll filling). Because there is evolutionary structure implicit in Gossypium with tetraploid species sharing very high homology to two diploid species and no other explanation is possible other than they merged to give rise to the tetraploids, the sequences are compared among these living species to infer similar gene functions among homologous genetic sequences. Higher order structures of genetic sequences such as amino acid codes, intron/exon sizes and patterns, protein structure and activity and locations on chromosomes and activity in nucleus and expression in specific tissues are further indications of their likely roles as phosphatases in ABA pathways. As long as the experimental data continue to align, then the hypothesis that these are putative sequences in PP2CAs and ABA pathways becomes more likely. Also Interactions are expected and are tested in yeast two way hybrids and are further indications of function, homology and likely evolutionary relationships among these species. The tetraploid nature of G. hirsutum and G. barbadense and even selection pressures by man to produce cotton in agricultural settings are expected to create additional variation in sequences and expression, measured further with tissue specific assays.

Some statement(s) like these or some rationale before each description of each method will greatly assist the readers in following the purpose and hypotheses specific to each study. There is a lot of information and evidence gathering but the reader still needs to know the significance of each. Perhaps citations to another article that used the same method would suffice for brevity.

Experimental design

I appreciate the thorough collection of the data, but want more information on details of each, and how the output is generated, analyzed and fed into the next method/step. It was assumed that the readers all know the steps, but if that is the case, your audience becomes very small. Some sentences with each method can help understand the expected results or the hypotheses and help the reader to try to repeat your results. The variety of methods and data output is acceptable to me and to the investigative goals of the article. I have confidence that the findings could lead to more research on PP2CAs and ABA in cotton.



Rigorous investigation performed to a high technical & ethical standard.

Give more detail on how you review the results of each method to assure quality for the next method.
The methods in this article are all common to this type of research and the data was well applied to each step and to the conclusions of the research.


Methods described with sufficient detail & information to replicate.

More detail is needed because so many methods, programs, sequences are utilized in this study. There are hundreds, or thousands of other lines of the four species that you used that could also be analyzed. There are 40 some other species of Gossypium that could also be used in this type of study. Ask yourself, how would another scientist repeat every step of your study in order to contribute the same useful data and make similar conclusions about PP2CAs and ABA pathways in Gossypium? Imagine if I stated my research as, planted Gossypium trilobum, harvested leaves, extracted DNA, ran AFLP and created a linkage map. Obviously, there is a lot of detail to each step. It seems to be an easy fix as you have plenty of methods but the readers need more details and then can be inserted or clarified.

One example where more detail is needed is:
Line 153 Measurements of PP2CAs expression in tissues and in response to ABA or osmotic stress
It indicates TM-1 plants, but how many plants were grown? It also indicates flowers and ovules were harvested, but how many flowers were harvested? At least tell the readers if there is a quantitative product needed like so many ug/ml of cDNA and it would indicate that say 50 flowers were harvested to generate at least 2 mL of a 20 ug/ml concentration of cDNA, something to help the readers understand and replicate this study. It ends with saying it was replicated three times, but this is vague. Was it done with different plants to avoid injury effects of resampling? Was it done a week apart, a month apart? Plant expression, especially with ABA can be very sensitive and even forgetting to water the plants one day could alter the results so the details are important. How was the PEG6000 applied, into the MS media or in a spray? I am assuming that this replication was how you got the standard errors, but what were the actual values put in and the results of ANOVA? Even though you cite some previous articles so they can read up on the methodology the readers still need some detail to visualize your work and relate it to their methods. Many readers will be learning this for the first time, or at least the first time with respect to cotton, so the more detail the more helpful. Results from one are funneled into another method so how is the quality control assured in each step, or the cutoff values to selecting sequences for the next step? For example, the results from a BLAST search can be longer than an encyclopedia set so how was it trimmed down to a best set of sequences to continue?

Validity of the findings

The novelty of the research makes it difficult to determine validity. Be aware that more proofs are needed before one is 100% sure that a given sequence is a specific enzyme. The wealth of methods and data give strength to your findings and this should be emphasized.

How were the standard errors generated? The consistency of putative PP2CA sequences throughout all of your methods gives weight to the data.

The conclusions are conservative but appropriate. Meaningful data was generated and presented. I would explain more about figure 3, because I see it in many places but not much is concluded. It is pretty but I am not sure what conclusions to draw from it.

Emphasize the collinearity among A chromosomes 1 – 13 (G. arboreum) and D (G. raimondii) and At and Dt because it is central to all this research, and it ties decades of classical genetics, breeding and cytology into evolution of the Gossypium genus. It is a contributing reason you went so quickly from just genomic sequences to finding a class of genetic sequences for function as PP2CAs. It also helps to show where is the genetic diversity, how it was formed, and how it can be dissected and better understood and utilized. I am curious about the brief mention of ‘seed germination’ in your introduction because maybe these pathways have an effect in my work with exotic cotton germplasm. You have a very good introduction that provided a information on these pathways so how much more do we know from your research, because it is not always obvious to the readers. The conclusion should list the ways it can spawn new research especially if you think it is too early to speculate too much on the results.

It seems to be an article that opens many doors to future research. If more speculation is desired than an example would be that the separation of PP2CAs according to time and tissue and stress responses indicates separate loci amenable to selection by cotton breeders for trait improvement or by cotton scientists for research into complex pathways in Gossypium. Extension of these experiments to wild species or other cotton pedigrees will further dissect the regulation and genetics of PP2CAs and the role of PP2CAs and ABA in cotton growth and productivity.

Additional comments

It is good research because it starts with mere genomic sequences and then ends up with putative genes and means to dissect the evolution and physiology of Gossypium. Make it easier for another researcher to duplicate the research so that more knowledge can be added.

Reviewer 2 ·

Basic reporting

As a pivotal phytohormone under water-limited conditions, abscisic acid (ABA) has been intensively studied in model plants such as Arabidopsis thaliana, and its signaling has been well characterized at molecular levels. In this manuscript, the authors examined one of the important players of ABA signaling, type 2C protein phosphatases (PP2Cs) classified in clade A, in four cotton (Gossypium) species. This study might provide interesting insights into ABA signaling in fiber crops, and the results were presented in a standard manner. However, there were several concerns.

1-1. The authors should reconsider designation of the names of PP2Cs and the protein subfamily. First, I disagree with the designation of Gossypium clade A PP2Cs named after the Arabidopsis PP2Cs (Table 1; line 183-186). Because the nine Arabidopsis PP2Cs were not one-to-one orthologs of the Gossypium PP2Cs according to the phylogenetic relationships (Fig. S1-S4), it should not be objective. In addition, all of the nine Arabidopsis PP2Cs have been named based on the characteristic of genetics and transcripts, such as ABA insensitive 1 (ABI1). If the authors used the same terms for Gossypium, readers should be confused. I would suggest simply numbering the PP2Cs as the previous report (Xue et al., 2008) cited in this manuscript.
Secondly, to avoid confusion, the abbreviation of clade A PP2Cs, PP2CAs, should not be used because one well-characterized clade A PP2C in Arabidopsis has been named as PP2CA (Tähtiharju and Palva, 2001), the alias of which is AHG3 (Yoshida et al., 2006).

References: Tähtiharju, Sari, and Tapio Palva. "Antisense inhibition of protein phosphatase 2C accelerates cold acclimation in Arabidopsis thaliana." The Plant Journal 26.4 (2001): 461-470.
Yoshida, Tomo, et al. "ABA-hypersensitive germination3 encodes a protein phosphatase 2C (AtPP2CA) that strongly regulates abscisic acid signaling during germination among Arabidopsis protein phosphatase 2Cs." Plant physiology 140.1 (2006): 115-126.

1-2. Please cite the original studies for the contexts (lines 82-84 and 90-92). The followings should be more appropriate.

References: Fujii, Hiroaki, and Jian-Kang Zhu. "Arabidopsis mutant deficient in 3 abscisic acid-activated protein kinases reveals critical roles in growth, reproduction, and stress." Proceedings of the National Academy of Sciences 106.20 (2009): 8380-8385.
Fujii, Hiroaki, et al. "In vitro reconstitution of an abscisic acid signalling pathway." Nature 462.7273 (2009): 660.
Fujita, Yasunari, et al. "Three SnRK2 protein kinases are the main positive regulators of abscisic acid signaling in response to water stress in Arabidopsis." Plant and Cell Physiology 50.12 (2009): 2123-2132.
Geiger, Dietmar, et al. "Activity of guard cell anion channel SLAC1 is controlled by drought-stress signaling kinase-phosphatase pair." Proceedings of the National Academy of Sciences 106.50 (2009): 21425-21430.
Lee, Sung Chul, et al. "A protein kinase-phosphatase pair interacts with an ion channel to regulate ABA signaling in plant guard cells." Proceedings of the National Academy of Sciences 106.50 (2009): 21419-21424.

1-3. Likewise, the references seemed to be incorrect (line 315). It should be Ma et al., 2009 and Park et al., 2009, as cited in the Introduction (lines 86-87).

Experimental design

The authors sought PP2Cs genes in four cotton species, and performed phylogenetic and synteny analyses especially focusing on clade A PP2Cs. There were major concerns on prediction of subcellular localization (Table 1) and protein-protein interaction assays (Figures 9-11).

2-1. As described in the Introduction (lines 79-82), it has been shown that Arabidopsis clade A PP2Cs interact with a number of cytosolic and nuclear proteins. It seemed to be inconsistent with the prediction that most of Gossypium clade A PP2Cs are localized in nucleus (Table 1; lines 192-195). Have the authors verified the accuracy of prediction (Plant-mPLoc: http://www.csbio.sjtu.edu.cn/bioinf/plant-multi/; lines 124-125)? If Arabidopsis clade A PP2Cs are predicted to be localized in different compartments in contrast to literature knowledge and other prediction database (such as SUBA; http://suba.live/), the accuracy of Plant-mPLoc should be carefully verified. Alternatively, since it is just a prediction and not experimentally validated, the result can be excluded from Table 1.
Moreover, the notion would not be rational (lines 345-347). As described above, Arabidopsis clade A PP2Cs are thought to be present in cytosol and nucleus to dephosphorylate plasma membrane proteins and transcription factors. Given that subcellular localization of rice PP2Cs have not been reported in one cited paper (Xue et al., 2008), the sentence was literally wrong.

2-2. The result (Figures 9-11) should be cautiously interpreted. Given that the homology between GhPYLs and AtPYLs was not supported by any data and references, except for the accession numbers (lines 317-318), it was unclear if these GhPYLs have roles as ABA receptors. To argue that they ‘may play roles via interaction with GhPYLs in ABA-dependent or ABA-independent manner in cotton’ (lines 380-381), further supporting data showing that GhPYLs are orthologs of AtPYLs (for example, alignment of protein sequences) are requested.
Meanwhile, the result did not indicate that they are ‘functional phosphatases’ (line 380). Any biochemical evidence are needed.

Validity of the findings

More rational interpretation and further discussion would be required.

3-1. The authors should interpret the result objectively (lines 188-192, 343-344). I would not think that most of the clade A PP2Cs in Gossypium has ‘similar’ properties, because of the varieties in protein size and pI.

3-2. I would feel that most of the text in the Discussion section were already described in the Result section. For example, the paragraph (lines 367-370) was a repetition of the result (lines 274-276). More references should be requested for discussion and interpretation of the results.

Additional comments

4-1. Abstract: Please precisely describe the biological role of PYL genes. Otherwise, readers could not tell the importance of the result of yeast two-hybrid assays (lines 45-47).

4-2. Please consider revising the description (lines 69-70). Clade A PP2Cs are the ones of well-studied PP2Cs in Arabidopsis, and they have been shown to have important roles in ABA signaling. But, nobody knows if they are the most important in plants.

4-3. The statement should be incorrect (lines 75-77). Not only ABI1 and ABI2, HAB1 and AHG3/PP2CA have been well studied and these PP2Cs are thought to cooperatively function in many signaling events.

Reference: Rubio, Silvia, et al. "Triple loss of function of protein phosphatases type 2C leads to partial constitutive response to endogenous abscisic acid." Plant physiology 150.3 (2009): 1345-1355.

4-4. A minor technical comment. The authors showed protein-protein interaction by yeast two-hybrid assay. Although no information was given in the legends, I assumed that black triangles indicate gradients of yeast culture. The gradients seemed not to be correlated with the spots in most cases, and thus I was wondering if the experiments were performed in a reproducible manner and if it is necessary to show all of the results as main figures.

4-5. Not supported by any experimental evidence (lines 386-387). Please tone down the description.

4-6. The phylogenetic relationships between Gossypium species and Arabidopsis were shown in Supplemental Figures (S1-S4). I would suggest using the same color code for the four figures. It would be much easier for readers to interpret the result if the PP2Cs of the same clade are shaded by the same color in the four figures.

4-7. Please specify the abbreviation ‘ABI’ (line 73).

4-8. It should be ‘Highly ABA-induced PP2C 1’ (line 74).

4-9. References are needed (lines 75-77, 79-80, and 80-82).

4-10.What does ‘structure’ mean (line 95)? It should be ‘domain structure’.

4-11. There were a couple of typographical errors.

---

## Round 0.2 · Minor Revisions

Both re-reviews of your revised manuscript were positive and I feel the manuscript will be acceptable for publication if you complete the revisions highlighted in the reviews.

I found one additional minor issue I'd like you to correct and I have one other request regarding the phylogenetic analyses. First, the correction:

Lines 206-208 currently reads: "The ModelFinder, tree reconstruction and ultrafast bootstrap were used (Minh et al., 2013). The evolutionary tree was redrawn by the FigTree v1.4.4 software."

Rewrite as: "The best-fitting model was chosen using ModelFinder (Kalyaanamoorthy et al. 2017) and support was assessed using the ultrafast bootstrap (Minh et al., 2013). Evolutionary tree were visualized using FigTree v1.4.4 (available from https://github.com/rambaut/figtree/releases)."

(Also add citation for modelfinder -- Kalyaanamoorthy, S., Minh, B. Q., Wong, T. K., von Haeseler, A., & Jermiin, L. S. (2017). ModelFinder: fast model selection for accurate phylogenetic estimates. Nature methods, 14:587.)

The other thing I would like you to make a file of your multiple sequence alignment files and your trees to add as supporting material. This could be done in two ways:

1. Collect all alignment files and all of the newick treefiles (for IQ-TREE these should have the suffix .treefile) into a single folder, name the files clearly, add a README file listing the filename and which tree it corresponds to, and provide a compressed version of that file as supporting data.

2. Make a folder as in part 1 but upload those data to a repository like https://zenodo.org/ and then provide information for people to download the repository (zenodo provides a full citation when you upload).

I don't think the revisions should take too long and I think the paper is acceptable if you make the revisions.

·

Basic reporting

Be more consistent in the use of italics. I am not sure if annotating the PP2CA with Gh, Gb, Ga, or Gr makes it all italics or should just the Gh be italicizeds. In appeared that the whole annotated sequence is italicized and I tried to correct it in the word doc. Doublecheck my changes and make sure you are consistent with the use of italics.
Changes should be in Orange color.

Experimental design

I am still unclear on the use of plants one experiment. It appeared that you were able to start sampling for all leaves, roots, stem and flowers just after 21 days. We know that a cotton plant is still seedling at 21 days. Can you be more clear on the age/size of the plant when you harvested the relevant tissue. It is not conceivable to me that a 21 day old plant can provide flowers for your experiment.

You did supplement better this time with all the materials and methods and citations.

Validity of the findings

I tend to enjoy the speculation part of the discussion and wrote my comments into the manuscript. Check that you agree with my speculation because you are the expert with your research and I am just a first time reader of it. I think you could say more about the Ka/Ks ratios as it could be significant for plants that are selected in nature versus selected by man. Remember you are assigning gene function on the basis of a number of indirect methods so stress the consistency of the many methods to the validity of assigning PP2CA gene function to these sequences.

Additional comments

It is a much improved manuscript this time around.
The additional edits are in orange in the pdf (still contain tracking from the previous edits).

Reviewer 2 ·

Basic reporting

Abscisic acid (ABA) plays important roles in plant abiotic stress responses, and its signaling event has been characterized in the model plant Arabidopsis thaliana. To gain insights into molecular mechanisms of stress responses in cotton, the authors studied type 2C protein phosphatases (PP2Cs), which are crucial components in ABA perception and signal transduction, in four cotton species. The results were concisely presented, and the concerns raised by previous reviewers have been addressed.

There were several minor concerns.

1. The tissue-specific expression of GhPP2CA was presented in Figure 6A and B. It might be inconsistent with the text (lines 290-291), because GhPP2CA27 (Figure 6B) and other GhPP2CAs were highly expressed in fibers. Please consider to include others in Figure 6B or omit classification ‘A’ and ‘B’.

2. The following sentence was incorrect (lines 390-391): GhPYLs are homologs of AtPYLs and some GhPYLs are functional ABA receptors (Liang et al., 2017; Zhang et al., 2017b).
These studies did not provide any direct evidence that GhPYLs perceive ABA molecules. ‘Some GhPYLs have been suggested to be functional ABA receptors’ by these studies.

3. How many PP2CAs in Arabidopsis? It would be nine in Introduction (line 74), whilst it was ten in Discussion (line 337).

4. I would suggest spelling out ‘PP2CA’ at the first appearance in the text (line 70), except for Abstract.

5. Line 32: negatively?

6. Lines 279 and 282: cacao?

7. Line 299: ‘GhPP2CA2-4, 8, 10, 24’?

Experimental design

no comment

Validity of the findings

no comment

---

## Round 0.3 · accepted · Accept

Thanks for completing the requested edits. I've looked through and feel you have addressed everything. I look forward to seeing your manuscript in press!